# On the Convergence of SVGD in KL divergence via Approximate gradient flow

**Masahiro Fujisawa**[*]  *fujisawa@ist.osaka-u.ac.jp*
*The University of Osaka*
*RIKEN Center for Advanced Intelligence Project*
*Lattice Lab, Toyota Motor Corporation*

**Futoshi Futami**[*]  *futami.futoshi.es@osaka-u.ac.jp*
*The University of Osaka*
*RIKEN Center for Advanced Intelligence Project*

**Reviewed on OpenReview:** *https://openreview.net/forum?id=AG1zXt5aoA*

## Abstract

This study investigates the convergence of Stein variational gradient descent (SVGD), which is used to approximate a target distribution based on a gradient flow on the space of probability distributions. The existing studies mainly focus on the convergence in the kernel Stein discrepancy, which doesn't imply weak convergence in many practical settings. To address this issue, we propose to introduce a novel analytical approach called $(\epsilon, \delta)$-*approximate gradient flow*, extending conventional concepts of approximation error for the Wasserstein gradient. With this approach, we show the sub-linear convergence of SVGD in Kullback–Leibler divergence under the discrete-time and infinite particle settings. Finally, we validate our theoretical findings through several numerical experiments. The code to reproduce our experiments is available at https://github.com/msfuji0211/svgd_convergence.

## 1 Introduction

Sampling from an unnormalized target distribution, such as posterior distribution in Bayesian inference, is a fundamental problem in machine learning. The mainstream approaches for obtaining such samples is using Markov Chain Monte Carlo (MCMC) methods (Hastings, 1970; Welling & Teh, 2011) or approximating the target distribution by variational inference (VI) (Jordan et al., 1999; Blei et al., 2017). While MCMC provides guarantees of producing asymptotically unbiased samples from the target density, it tends to be computationally intensive (Robert & Casella, 2004). On the other hand, VI achieves a computationally efficient approximation of the target distribution through stochastic optimization under a simpler alternative distribution; however, it does not come with a guarantee of obtaining unbiased samples (Blei et al., 2017).

To alleviate such sample bias while maintaining computational efficiency of VI as much as possible, Liu & Wang (2016) introduced *Stein variational gradient descent* (SVGD), which allows the direct approximation of the target distribution without the need for alternative distributions. SVGD iteratively updates correlated samples, referred to as *particles*, by minimizing the Kullback–Leibler (KL) divergence between a distribution of particles and the target distribution through a gradient flow on the space of probability distributions. Since the Wasserstein gradient is intractable in practice, SVGD approximates it through a kernel method.

On the theoretical front, analysis has been actively conducted ever since Liu (2017) elucidated the asymptotic behavior of SVGD from the perspective of gradient flow within the reproducing kernel Hilbert space (RKHS). Korba et al. (2020) showed sub-linear convergence in kernel Stein discrepancy (KSD) under infinite particles assuming that KSD at each step is bounded. Salim et al. (2022) contributed a proof of sub-linear

---

[*]Equal contribution.

convergence in KSD without the necessity of bounded KSD assuming that the target distribution satisfies $T_1$ inequality (Villani, 2008), and Sun et al. (2023) provided the proofs of this convergence property by relaxing the smoothness assumption of the target distribution. A common thread in these analyses is seeing SVGD's update rule as the approximation of the Wasserstein gradient in the RKHS and showing that the KL divergence to target distribution monotonically decreases like gradient descent. Beyond the infinite particle setting, Shi & Mackey (2023) has recently shown that the SVGD with $n$ finite particles and an appropriate step size converges in KSD at the $\mathcal{O}(1/\sqrt{\log \log n})$ order if the target distribution is sub-Gaussian with a Lipschitz score.

However, the convergence analysis in terms of KSD is insufficient to understand the weak convergence property of SVGD because the convergence in KSD holds under highly restrictive conditions for the kernel and the target distribution under practical settings as shown by Gorham & Mackey (2017). This fact underscores the importance of conducting convergence analysis using criteria other than KSD to provide more realistic guarantees for the obtained particles. A natural candidate for the criterion is the KL divergence itself, which is the objective function of SVGD. Recently, Liu et al. (2023) showed that SVGD with finite particles achieves linear convergence in KL divergence under a very limited setting where the target distribution is Gaussian. However, the analytical approach presented in previous studies makes it difficult to conduct convergence analysis based on KL divergence in a more global setting. The reason for this lies in the fact that while the logarithmic Sobolev inequality (LSI) (Gross, 1975) is typically employed to show the linear convergence in KL divergence for a gradient flow in the space of probability distributions (Villani, 2008), it becomes apparent that the inequality similar to the LSI (see Eq. (7)) does not hold in practical settings (Duncan et al., 2023) when considering SVGD as a gradient flow in the RKHS.

In this study, we introduce a novel analytical approach that allows us to circumvent the aforementioned issue. A key idea in our analysis is to consider SVGD as an *approximation* of the gradient flow in the space of probability distributions, as opposed to the conventional analytical approach that views SVGD as a gradient flow in the RKHS. To express the degree of this approximation, we introduce a new concept called $(\epsilon, \delta)$-*approximate gradient flow*, which extends the concept of approximation error widely used in the gradient estimation context such as score gradient estimation (Lee et al., 2022; 2023) and particle-based VI (Liu et al., 2019; Dong et al., 2022).

With our concept, we offer new insights into the convergence of SVGD in the settings of discrete-time and an infinite number of particles. We first analyze the degree of the approximation error $\{\epsilon, \delta\}$ between the Wasserstein gradient of the KL divergence and the update rule in SVGD by focusing on spectral decomposition specified via a kernel function. With this approximation error analysis, we show that SVGD exhibits sub-linear convergence in the KL divergence for the first time, to the best of our knowledge. At last, we conduct a numerical study to examine the convergence behavior of SVGD across various metrics and validate the soundness of our theoretical findings.

## 2 Preliminaries

Random variables are denoted by capital letters like $X$, while deterministic values are denoted by lowercase letters like $x$. The Euclidean inner product and distance are expressed as $\langle \cdot, \cdot \rangle$ and $\| \cdot \|$, respectively. Let $\mathcal{X} = \mathbb{R}^d$ and let $C^l(\mathcal{X}, \mathcal{Y})$ be the space of $l$ continuously differentiable functions from $\mathcal{X}$ to a Hilbert space $\mathcal{Y}$. We abbreviate $C^l(\mathcal{X}, \mathbb{R})$ as $C^l(\mathcal{X})$. The set of smooth functions with compact support is expressed as $C_c^\infty(\mathcal{X})$. If $\phi \in C^1(\mathcal{X})$, its gradient is $\nabla \phi$. For $\phi \in C^1(\mathcal{X}, \mathcal{X})$, the Jacobian is represented as $J\phi(x)$, a $d \times d$ matrix at each point $x \in \mathcal{X}$. We define $\text{div}\phi(x) = \text{Tr}J\phi(x)$. The Hilbert–Schmidt and operator norm of a matrix are denoted as $\| \cdot \|_{\text{HS}}$ and $\| \cdot \|_{\text{op}}$.

### 2.1 Wasserstein space and continuity equation

Here we summarize some of the basics of optimal transport that underlie our analysis. We denote the set of probability measures on $\mathcal{X}$ with finite second moments as $\mathcal{P}_2(\mathcal{X})$. For any $\mu \in \mathcal{P}_2(\mathcal{X})$, we express the set of measurable functions $f : \mathcal{X} \to \mathcal{X}$ with $\int \|f\|^2 d\mu < \infty$ as $L^2(\mu)$, with its norm and inner product as $\| \cdot \|_{L^2(\mu)}$ and $\langle \cdot, \cdot \rangle_{L^2(\mu)}$. Given a measurable map $T : \mathcal{X} \to \mathcal{X}$ and $\mu$, we denote the pushforward

measure of $\mu$ by $T$ as $T\#\mu \in \mathcal{P}_2(\mathcal{X})$, which is characterized by $\int \phi(T(x))\mathrm{d}\mu(x) = \int \phi(y)\mathrm{d}T\#\mu(y)$ for any measurable and bounded function $\phi$. Given $\mu, \nu \in \mathcal{P}_2(\mathcal{X})$, the Wasserstein distance between $\mu$ and $\nu$ is defined as $W_2^2(\mu, \nu) = \inf_{s \in \mathcal{S}(\mu,\nu)} \int \|x - y\|^2 \mathrm{d}s(x, y)$, where $\mathcal{S}(\mu, \nu)$ is the set of couplings between $\mu$ and $\nu$. This distance defines a metric on $\mathcal{P}_2(\mathcal{X})$, making $(\mathcal{P}_2(\mathcal{X}), W_2)$ the Wasserstein space, which is complete and separable.

Now we introduce a *continuous equation*. Let $T > 0$ and consider a weakly continuous map $\mu : (0, T) \to \mathcal{P}_2(\mathcal{X})$, $t \mapsto \mu_t$. The family $(\mu_t)_{t \in (0,T)}$ satisfies a continuity equation if there exists $(v_t)_{t \in (0,T)}$ such that $v_t \in L^2(\mu_t)$ and $\frac{\partial \mu_t}{\partial t} + \mathrm{div}(\mu_t v_t) = 0$ holds in the distribution sense (see Appendix B.1 for the formal meaning of *distribution sense*). A family $(\mu_t)_{t \in (0,T)}$ that satisfies a continuity equation with integrable $\|v_t\|_{L_2(\mu_t)}$ over $(0, T)$ is referred to as *absolutely continuous*. Conversely, one can construct an absolutely continuous $(\mu_t)_{t \in (0,T)}$ by selecting $(v_t)_{t \in (0,T)}$ such that they meet the above condition.

While the Wasserstein space does not inherently possess the characteristics of a Riemannian manifold, it can be endowed with a Riemannian structure and interpretation (Otto, 2001). In this interpretation, the tangent space of $\mathcal{P}_2(\mathcal{X})$ at $\mu_t$, denoted as $\mathcal{T}_{\mu_t}\mathcal{P}_2(\mathcal{X})$, forms a subset of $L^2(\mu_t)$. When considering all possible $(v_t)_{t \in (0,T)}$, we call $v_t$ that exhibits the minimal $L^2(\mu_t)$ norm as the velocity field of $(\mu_t)_{t \in (0,T)}$. This minimality condition can be characterized by the requirement that $v_t \in \mathcal{T}_{\mu_t}\mathcal{P}_2(\mathcal{X})(\subset L^2(\mu_t))$.

## 2.2 Sampling-based Approximation via Gradient flow of KL divergence

We aim to obtain samples from the density $\pi(x) \propto e^{-V(x)}$ in $\mathcal{P}_2(\mathcal{X})$ under the following assumption for the potential function $V : \mathcal{X} \to \mathbb{R}$.

**Assumption 1.** *The Hessian of $V \in C^2(\mathcal{X})$, $H_V$, satisfies $\|H_V\|_{\mathrm{op}} \leq L$.*

This task can be formulated as the optimization problem over a functional space, i.e., minimizing a functional, KL divergence of $\mu$ from $\pi$ defined on Wasserstein space, that is,

$$\min_{\mu \in \mathcal{P}_2(\mathcal{X})} \mathrm{KL}(\mu|\pi), \quad \mathrm{KL}(\mu|\pi) \coloneqq \int \log \frac{\mathrm{d}\mu}{\mathrm{d}\pi}(x)\mathrm{d}\mu(x), \tag{1}$$

where $\mathrm{KL}(\cdot|\pi) : \mathcal{P}_2(\mathcal{X}) \to [0, +\infty)$, $\mu \mapsto \mathrm{KL}(\mu|\pi)$ and $\mu$ is absolutely continuous with respect to (with respect to) $\pi$. Thus, Radon–Nikodym [1] derivative $\mathrm{d}\mu/\mathrm{d}\pi$ is available ($\mathrm{KL}(\mu|\pi) = +\infty$ otherwise).

As a method for solving Eq. (1), a gradient-descent-like algorithm utilizing the differential structure of the Wasserstein space and continuous equations (see Section 2.1) is often employed. Let the Wasserstein gradient of $\mathrm{KL}(\mu|\pi)$ at $\mu$ be $\nabla_{W_2}\mathrm{KL}(\mu|\pi)$ (the formal definition is presented in Appendix B.1). We then consider how $\mathrm{KL}(\mu|\pi)$ evolves by the continuity equation, i.e.,

$$\frac{\mathrm{d}}{\mathrm{d}t}\mathrm{KL}(\mu_t|\pi) = \langle \nabla_{W_2}\mathrm{KL}(\mu_t|\pi), v_t \rangle_{L^2(\mu_t)}, \tag{2}$$

which shows that $\mathrm{KL}(\mu|\pi)$ is minimized by choosing $v_t$ such that $\langle \nabla_{W_2}\mathrm{KL}(\mu_t|\pi), v_t \rangle_{L^2(\mu_t)} \leq 0$ and using the continuity equation. A natural choice is to use the Wasserstein gradient itself as $v_t = -\nabla_{W_2}\mathrm{KL}(\mu_t|\pi)$, which results in $\frac{\mathrm{d}}{\mathrm{d}t}\mathrm{KL}(\mu|\pi) = -\|\nabla_{W_2}\mathrm{KL}(\mu_t|\pi)\|_{L^2(\mu_t)}^2 \leq 0$. According to the fact that the Wasserstein gradient of KL divergence is obtained as $\nabla_{W_2}\mathrm{KL}(\mu|\pi) = \nabla \log \frac{\mu}{\pi} \in L^2(\mu)$ (Ambrosio et al., 2005), we have

$$\frac{\mathrm{d}}{\mathrm{d}t}\mathrm{KL}(\mu_t|\pi) = -\left\|\nabla \log \frac{\mu_t}{\pi}\right\|_{L^2(\mu_t)}^2. \tag{3}$$

Many existing studies analyzed Eq. (3) under the following assumption (Bakry et al., 2013).

**Assumption 2.** *We say that the target distribution $\pi$ satisfies the LSI, if, for any $\mu \in \mathcal{P}_2(\mathcal{X})$, there exists a positive constant $C_{\mathrm{LS}}$ such that*

$$\mathrm{KL}(\mu|\pi) \leq \frac{1}{C_{\mathrm{LS}}}\left\|\nabla \log \frac{\mu}{\pi}\right\|_{L^2(\mu)}^2. \tag{4}$$

---

[1] Suppose that $\mu$ is absolutely continuous with respect to $\pi$, i.e., $\mu \ll \pi$. Then, there exists a function $f$ such that, for any measurable set $A$, $\mu(A) = \int_A f(x)\mathrm{d}\pi(x)$. This function $f$ is referred to as the Radon–Nikodym derivative of $\mu$ with respect to $\pi$, denoted by $f = \mathrm{d}\mu/\mathrm{d}\pi$ (Durrett, 2019).

With the above inequality and Eq. (3), we have $\mathrm{KL}(\mu_t|\pi) \leq e^{-C_{\mathrm{LS}}t}\mathrm{KL}(\mu_0|\pi)$, which implies linear convergence. However, it is difficult to deal with the continuous-time equation of Eq. (3), and thus discretization such as a forward Euler discretization (Ambrosio et al., 2005) is often used. This recursion is given by

$$\mu_{t+1} = \left(I - \gamma_t \nabla \log \frac{\mu_t}{\pi}\right) \#\mu_t, \tag{5}$$

at each iteration $t$ [2], where $\gamma_t > 0$ is a stepsize and $I$ is the identity map.

## 2.3 Stein variational gradient descent

Performing optimization based on Eq. (5) is still difficult because $\mu$ is often intractable and thus $\nabla \log \frac{\mu}{\pi}$ is hard to compute. SVGD is one of the alternative gradient flow approaches to avoid this issue by projecting $\nabla \log \frac{\mu}{\pi}$ into the reproducing kernel Hilbert space (RKHS) by a kernel function.

Here, we briefly summarize the fundamental operations on the RKHS. Let $k : \mathcal{X} \times \mathcal{X} \to \mathbb{R}$ be a symmetric and positive semi-definite kernel and $\mathcal{H}_0$ be its corresponding RKHS of real-valued functions $\mathcal{X} \to \mathbb{R}$. The inner product within $\mathcal{H}_0$ is denoted as $\langle \cdot, \cdot \rangle_{\mathcal{H}_0}$, which satisfies $f(x) = \langle f, k(\cdot, x) \rangle_{\mathcal{H}_0} (\forall f \in \mathcal{H}_0)$ by the reproducing property of $\mathcal{H}_0$. We also define $\mathcal{H}$ as the Cartesian product of $\mathcal{H}_0$, whose elements are expressed as $f = (f_1, \ldots, f_d)$ where $f_i \in \mathcal{H}_0$ for $i = 1, \ldots, d$. The inner product of $f, g \in \mathcal{H}$ is given by $\langle f, g \rangle_{\mathcal{H}} = \sum_{i=1}^d \langle f_i, g_i \rangle_{\mathcal{H}_0}$. If $\mu \in \mathcal{P}_2(\mathcal{X})$ and $\int k(x,x)\mathrm{d}\mu(x) < \infty$, the integral operator associated to $k$ and $\mu$ can be defined as $S_{\mu,k}f(x) := \int k(y,x)f(y)\mathrm{d}\mu(y)$, where $S_{\mu,k} : L^2(\mu) \to \mathcal{H}$ and thus $\mathcal{H} \subset L^2(\mu)$ [3]. We further define the inclusion map as $\iota : \mathcal{H} \to L^2(\mu)$, which is the adjoint of $S_{\mu,k}$. Under the map $\iota$, for $f \in L^2(\mu)$ and $g \in \mathcal{H}$, we have $\langle f, \iota g \rangle_{L^2(\mu)} = \langle \iota^* f, g \rangle_{\mathcal{H}} = \langle S_{\mu,k}f, g \rangle_{\mathcal{H}}$, where $\iota^*$ is the adjoint of $\iota$. We finally define the mapping function $P_{\mu,k} : L^2(\mu) \to L^2(\mu)$, where $P_{\mu,k} = \iota S_{\mu,k}$.

In SVGD, instead of using the Wasserstein gradient $\nabla \log \frac{\mu}{\pi}$, we employ $-P_{\mu,k} \nabla \log \frac{\mu}{\pi}$ as $v_t$ in Eq. (2), leading to the following discretized dynamics:

$$\mu_{t+1} = \left(I - \gamma_t P_{\mu,k} \nabla \log \frac{\mu_t}{\pi}\right) \#\mu_t. \tag{6}$$

The difference from Eq. (5) is that $\nabla \log \frac{\mu}{\pi}$ is mapped by $P_{\mu,k}$. If a kernel function satisfies $\lim_{\|x\| \to \infty} k(x, \cdot)\pi(x) = 0$, by using an integration by parts (Liu, 2017), we can obtain $P_{\mu,k} \nabla \log \frac{\mu}{\pi}(x) := -\int [\nabla \log \pi(y) k(y,x) + \nabla_y k(y,x)] \mathrm{d}\mu(y)$. By focusing on the continuous dynamics of the KL divergence, we have

$$\frac{\mathrm{d}}{\mathrm{d}t}\mathrm{KL}(\mu_t|\pi) = -\left\langle \nabla \log \frac{\mu_t}{\pi}, P_{\mu_t,k} \nabla \log \frac{\mu_t}{\pi} \right\rangle_{L^2(\mu_t)} = -\left\| S_{\mu_t,k} \nabla \log \frac{\mu_t}{\pi} \right\|_{\mathcal{H}}^2 =: -I_{\mathrm{stein}}(\mu_t|\pi),$$

where $I_{\mathrm{stein}}(\mu_t|\pi)$ is called as the Stein–Fisher (SF) information (Duncan et al., 2023). It is known that the square root of the SF information corresponds to the KSD. Now it is tempting to consider whether the inequality similar to LSI in Eq. (4) holds for the SF information presented below:

$$\mathrm{KL}(\mu|\pi) \leq c I_{\mathrm{stein}}(\mu|\pi), \tag{7}$$

where $c$ is some positive constant. If this inequality holds, the linear convergence of SVGD holds. Unfortunately, the conditions for the validity of this inequality are not as evident as in the case of LSI and Duncan et al. (2023) has shown that Eq. (7) may not hold in many practical models with kernel functions like the RBF kernel, where the tail of $\pi$ is exponential. Hence, showing the linear convergence of KL divergence in the geometry of $\mathcal{H}$ is not straightforward. We refer to Liu (2017) and Duncan et al. (2023) for a detailed discussion of the geometry of SVGD.

Recently, Salim et al. (2022) showed the descent lemma, $\mathrm{KL}(\mu_{t+1}|\pi) \leq \mathrm{KL}(\mu_t|\pi) - c\gamma I_{\mathrm{stein}}(\mu_t|\pi)$ holds where $c$ is some positive constant that depends on the problem. Although we can obtain the convergence in KSD from this inequality, the convergence KSD not necessarily means the weak convergence as discussed in Gorham & Mackey (2017).

---

[2] For the sake of readability, we adopt $t$ to express both continuous and discrete time.

[3] We introduce $S_{\mu,k}$ for vector inputs $f = (f_1, \ldots, f_d)$. When $f$ is a scalar ($d = 1$), for simplicity, we consider $S_{\mu,k}$ to be defined as applied to a single element, i.e., $S_{\mu,k} : L^2_0(\mu) \to \mathcal{H}_0$, allowing us to abuse the notation, where $L^2_0(\mu)$ is the set of a measurable function $f_1 : \mathcal{X} \to \mathbb{R}$ with $\int f_1^2 \mathrm{d}\mu < \infty$.

# 3 Approximate gradient flow

Here, we introduce a new concept of approximation for the Wasserstein gradient, $(\epsilon, \delta)$-*approximate gradient flows* (AGF). We then analyze the convergence of the KL divergence under our concept.

## 3.1 $(\epsilon, \delta)$-approximate gradient flow

Let us assume that a gradient flow on the Wasserstein space exists, which is induced by some velocity $v_t = g_{\mu_t}(x) \in L^2(\mu_t)$ for $x \in \mathcal{X}$. Here, $g_{\mu_t}(x)$ represents a function of $x$ only depending on $\mu_t$. In the continuous-time setting, such a gradient flow is obtained via the continuity equation given as $\frac{\partial \mu_t}{\partial t} + \mathrm{div}(\mu_t g_{\mu_t}(x)) = 0$. Under mild growth and regularity assumptions on $g_{\mu_t}(x)$ (Ambrosio et al., 2005; Bonnet & Frankowska, 2021), the existence and uniqueness of a gradient flow by $g_{\mu_t}$ is guaranteed. When considering discrete time, we assume that the recursion $\mu_{t+1} = (I - \gamma_t g_{\mu_t}) \# \mu_t$ exists, which is similar to Eq. (6).

In the presence of these, we consider the time evolution of $\mathrm{KL}(\mu_t|\pi)$ under the velocity $v_t = g_{\mu_t}(x)$ as in Section 2.2. In the continuous-time setting, we assume that $\frac{\mathrm{d}}{\mathrm{d}t}\mathrm{KL}(\mu_t|\pi) = \left\langle \nabla \log \frac{\mu_t}{\pi}, g_{\mu_t} \right\rangle_{L^2(\mu_t)}$. As for the discrete-time setting, we assume the following inequality with a kind of descent property:

$$\mathrm{KL}(\mu_{t+1}|\pi) \leq \mathrm{KL}(\mu_t|\pi) - \eta_t \left\langle \nabla \log \frac{\mu_t}{\pi}, g_{\mu_t} \right\rangle_{L^2(\mu_t)}, \tag{8}$$

where $\eta_t$ is some positive constant. Such a descent property holds both in the Wasserstein gradient flow (Ambrosio et al., 2005) and in SVGD as shown in Section 2.3.

From the above two (in)equalities, we can anticipate that when $g_{\mu_t}(x)$ exhibits behavior close to that of $\nabla \log \frac{\mu_t}{\pi}(x)$, i.e., $\left\langle \nabla \log \frac{\mu_t}{\pi}, g_{\mu_t} \right\rangle_{L^2(\mu_t)} \geq 0$ is satisfied (recall the cosine similarity in the finite-dimensional case), the KL divergence does not increase with $t$. In SVGD, for example, we set $g_{\mu_t} = P_{\mu_t,k} \nabla \log \frac{\mu_t}{\pi}$, which satisfies $\left\langle \nabla \log \frac{\mu_t}{\pi}, g_{\mu_t} \right\rangle_{L^2(\mu_t)} = I_{\mathrm{stein}}(\mu_t|\pi) \geq 0$.

However, the condition $\left\langle \nabla \log \frac{\mu_t}{\pi}, g_{\mu_t} \right\rangle_{L^2(\mu_t)} \geq 0$ is insufficient for explicitly analyzing the convergence rate since it doesn't convey how accurate the approximation via $g_{\mu_t}$ is. To overcome this situation, we introduce a new concept of the similarity between $\nabla \log \frac{\mu_t}{\pi}(x)$ and $g_{\mu_t}(x)$ as follows.

**Definition 1.** *Suppose that $\nabla \log \frac{\mu_t}{\pi}(x) < \infty$ (a.e.) and $\|\nabla \log \frac{\mu_t}{\pi}\|_{L^2(\mu_t)} < \infty$ for all $t$. Then, we say a function $g_{\mu_t}(x) \in L^2(\mu_t)$ is $(\epsilon_t, \delta_t)$-AGF if the following condition holds:*

$$-\left\langle \nabla \log \frac{\mu_t}{\pi}, g_{\mu_t} \right\rangle_{L^2(\mu_t)} \leq -\epsilon_t \left\| \nabla \log \frac{\mu_t}{\pi} \right\|^2_{L^2(\mu_t)} + \delta_t, \tag{9}$$

*where $\epsilon_t, \delta_t \geq 0$.*

Eq. (9) evaluates the approximation quality of $g_{\mu_t}(x)$ for $\nabla \log \frac{\mu_t}{\pi}$ via $\{\epsilon_t, \delta_t\}$, where $\epsilon_t$ and $\delta_t$ express the relative and absolute bias of approximating $\nabla \log \frac{\mu_t}{\pi}$ by $g_{\mu_t}(x)$, respectively. This definition is motivated by the inexact gradient descent methods in finite-dimensional parameter space such as (Jaggi, 2013; Schmidt et al., 2011) and unifies some existing approximate flow methods (see Section 3.2).

Using the $(\epsilon_t, \delta_t)$-AGF, we can analyze the convergence in KL divergence qualitatively as follows.

**Lemma 1.** *Suppose that Assumption 2 is satisfied. Then, under Eq. (8), for any $T \in \mathbb{N}$, we obtain $\mathrm{KL}(\mu_T|\pi) \leq \prod_{t=0}^{T-1}(1 - \eta_t \epsilon_t C_{\mathrm{LS}})\mathrm{KL}(\mu_0|\pi) + \sum_{t=0}^{T-1} \delta_t \eta_t \prod_{j=t+1}^{T-1}(1 - \eta_j \epsilon_j C_{\mathrm{LS}}).$*

*Proof.* By substituting Eq. (8) into Eq. (9) and applying the LSI, we obtain $\mathrm{KL}(\mu_{t+1}|\pi) \leq (1 - \eta_t \epsilon_t C_{\mathrm{LS}})\mathrm{KL}(\mu_t|\pi) + \eta_t \delta_t$. By induction in the above, we obtain the claim. $\qquad\qquad\square$

This lemma shows that $\epsilon_t$ and $\delta_t$ (as well as $\eta_t$) significantly impact the convergence rate.

**Remark 1.** *When $\delta_t = 0$ and $\eta_t \epsilon_t$ is independent of $t$, linear convergence is achieved, indicating that $g_{\mu_t}(x)$ provides a precise approximation of $\nabla \log \frac{\mu_t}{\pi}$. When $\delta = 0$ and $\eta_t \epsilon_t = \mathcal{O}(1/t^\alpha)$ with a constant $\alpha \in (0, 1]$, it indicates sub-linear convergence, which implies that the approximation quality is not so significant but it is enough to ensure the convergence in KL divergence.*

**Remark 2.** *If $\delta_t \neq 0$, the convergence is biased in terms of KL divergence. However, by employing the technique in Lee et al. (2022), it remains feasible to mitigate the impact of bias on total variation.*

### 3.2 Relation to existing approximate functional gradient flows

We now position our $(\epsilon, \delta)$-AGF framework as a general tool for analyzing a wide range of approximate functional gradient methods. We demonstrate that it not only provides a unified lens to interpret existing work but also clarifies the novelty of our analytical approach itself.

Dong et al. (2022) proposed the preconditioned functional gradient flow, where they considered approximating $\nabla \log \frac{\mu_t}{\pi}$ by neural networks (NNs). The authors also assumed that $g_{\mu_t}(x)$, which is the output of NNs, satisfies

$$\left\| g_{\mu_t} - \nabla \log \frac{\mu_t}{\pi} \right\|_{L^2(\mu_t)}^2 \leq \epsilon \left\| \nabla \log \frac{\mu_t}{\pi} \right\|_{L^2(\mu_t)}^2, \tag{10}$$

where $\epsilon < 1$. This corresponds to a special case of our AGF with $\epsilon_t := (1 - \epsilon)/2$ and $\delta_t := 0$, as confirmed by expanding the left-hand side of Eq. (10). According to Remark 1, the above inequality implies linear convergence in KL divergence. However, their method requires re-training NNs at each iteration, which yields difficulty in ensuring $\delta = 0$ in practice. Conversely, it later becomes evident that SVGD achieves $\delta_t = 0$ by using a kernel function that meets some conditions.

Lee et al. (2022; 2023) studied the score based diffusion models assuming that $g_{\mu_t}(x) = s(x) + \log \mu_t(x)$, where the $\nabla \log \pi(x)$ in the Wasserstein gradient is approximated with some measurable function $s(x)$ that satisfies

$$\left\| (s(\cdot) + \log \mu_t) - \nabla \log \frac{\mu_t}{\pi} \right\|_{L^2(\mu_t)}^2 \leq \delta. \tag{11}$$

The equation above corresponds to our AGF with $\epsilon_t = 0$, which signifies the presence of bias in the KL divergence (see Remark 2).

From the perspective of convergence analysis, the significant difference between these studies lies in the treatment of $\{\epsilon, \delta\}$. The convergence analysis in Dong et al. (2022), Lee et al. (2022), and Lee et al. (2023) assumes that $g_{\mu_t}$ achieves sufficiently small $\epsilon$ or $\delta$ according to the criteria in Eq. (10) or (11). In our study, we take the opposite approach — identifying $\{\epsilon, \delta\}$ that SVGD achieves under the AGF, and then evaluating its convergence properties.

Furthermore, this framework provides a structured path for future research. By decomposing the approximation error into $\epsilon_t$ that primarily governs the convergence rate and $\delta_t$ that introduces a final bias (as noted in Remarks 1 and 2), it allows for a more precise diagnosis and comparison of new and existing algorithms. This offers a more systematic methodology to guide future algorithmic improvements and theoretical analyses.

## 4 Application to Stein variational gradient descent

In this section, we present the main result, the convergence of SVGD in KL divergence, obtained by applying the concept of $(\epsilon, \delta)$-AGF, and provide an overview of the proofs. Here, $\mu_t$ represents the $t$-th output of the SVGD algorithm, where $t \in \mathbb{N}$ is the number of iterations as shown in Eq. (6).

### 4.1 Sub-linear convergence of SVGD in KL divergence

Our analyses are based on the following assumptions concerning the kernel function $k$.

**Assumption 3.** *The feature map $\nabla k(\cdot, x) : \mathcal{X} \to \mathcal{H}$ is continuous. Moreover, for all $x \in \mathcal{X}$, there exists $B > 0$ such that $\|k(\cdot, x)\|_{\mathcal{H}_0} \leq B$, $\sum_{i=1}^d \|\partial_i k(\cdot, x)\|_{\mathcal{H}_0}^2 \leq B^2$, and $\sum_{i,j=1}^d \|\partial_i \partial_j k(\cdot, x)\|_{\mathcal{H}_0} \leq B^2$ hold.*

**Assumption 4.** *The kernel $k$ is integrally strictly positive definite (ISPD), which means that $\int \int k(x, y) \mathrm{d}\rho(x) \mathrm{d}\rho(y) > 0$ holds for all finite nonzero signed Borel measures $\rho$.*

**Assumption 5.** *The trace of a kernel is bounded for any $\mu \in \mathcal{P}_2(\mathcal{X})$, i.e., $\int k(x, x) \mathrm{d}\mu(x) < \infty$.*

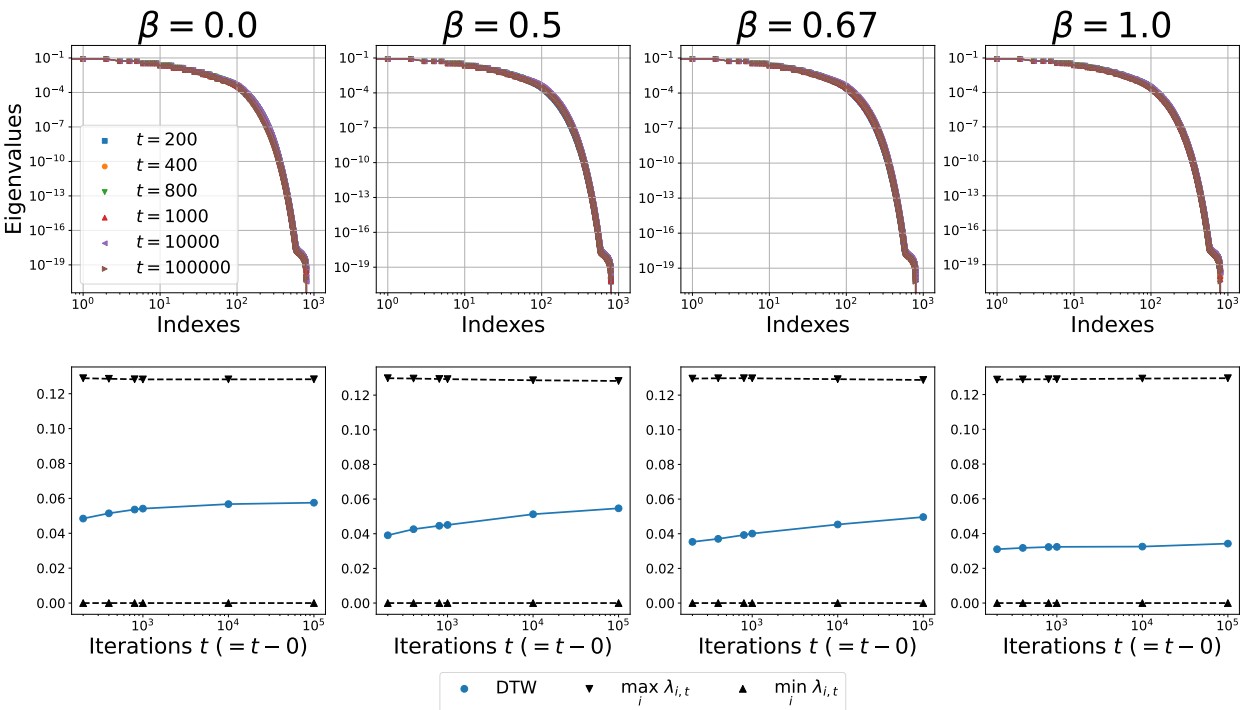

Figure 1: Behavior of eigenvalues $\boldsymbol{\lambda}_t = \{\lambda_{i,t}\}_{i=1}^m$ (first row) and the dynamic time warping (DTW) (second row) when we use the RBF kernel for SVGD with 1000 particles ($m = 1000$) in the multivariate normal experiments, where the symbols $m$ and $t$ represent the number of particles and the (discrete) time steps, respectively. The term $\beta$ corresponds to the factor of learning rate decay (see Section 5 for the details of the experimental settings). The eigenvalues were measured in increments of 100 for every iteration. We show the results up to the point where the KL divergence has not converged ($t = 200$ to $10^3$), as well as the results after convergence ($t = 10^4, 10^5$) according to Figure 4.

Under Assumption 5, the Hilbert–Schmidt operator $P_{\mu,k}$ has positive eigenvalues $\{\lambda_i\}$ (see Section 4.2 and Appendix C). We thus further pose the following assumption according to this fact.

**Assumption 6.** *Eigenvalues $\{\lambda_i\}$ are constant order with respect to $t$ and strictly positive, i.e., there exist upper and lower bounds for $\{\lambda_i\}$ that are independent of $t$ and are greater than $0$.*

Assumptions 3-6 are satisfied in the RBF kernel commonly employed in SVGD. A detailed discussion of these assumptions can be found in Appendix A.

We now show the main contribution of this paper, which establishes the sub-linear convergence of SVGD in KL divergence.

**Theorem 1.** *Suppose that Assumptions 1-6 are satisfied. Let $\alpha > 1$ and the stepsize $\gamma_t$ satisfies $\gamma_t \leq \mathcal{O}(1/t^{2/3})$ and $\gamma_t \leq (\alpha - 1)\alpha B^2 (1 + \|\nabla V(0) + L\mathbb{E}_\pi \|x\| + L\sqrt{2C_{\mathrm{LS}}^{-1}\mathrm{KL}(\mu_0|\pi)})(=: C_\gamma)$ for all $t$. Then, SVGD is $(c_0, 0)$-AGF and for any $T \in \mathbb{N}$, we have*

$$\mathrm{KL}(\mu_T|\pi) \leq \prod_{t=0}^{T-1} (1 - c_0\gamma_t)\,\mathrm{KL}(\mu_0|\pi), \tag{12}$$

*where $c_0(> 0)$ is a problem-dependent constant that is independent of $t$.*

This theorem guarantees the sub-linear convergence of SVGD in KL divergence because $\lim_{t\to\infty} \frac{\mathrm{KL}(\mu_{t+1}|\pi)}{\mathrm{KL}(\mu_t|\pi)} = \lim_{t\to\infty} 1 - c_0\gamma_t = 1$. Moreover, by setting $\gamma_t = \frac{c_1}{t}$ for some positive constant $c_1$ in the above, for example, we obtain $\mathrm{KL}(\mu_T|\pi) \leq \frac{\mathrm{KL}(\mu_0|\pi)}{T^{c_1 c_0}}$.

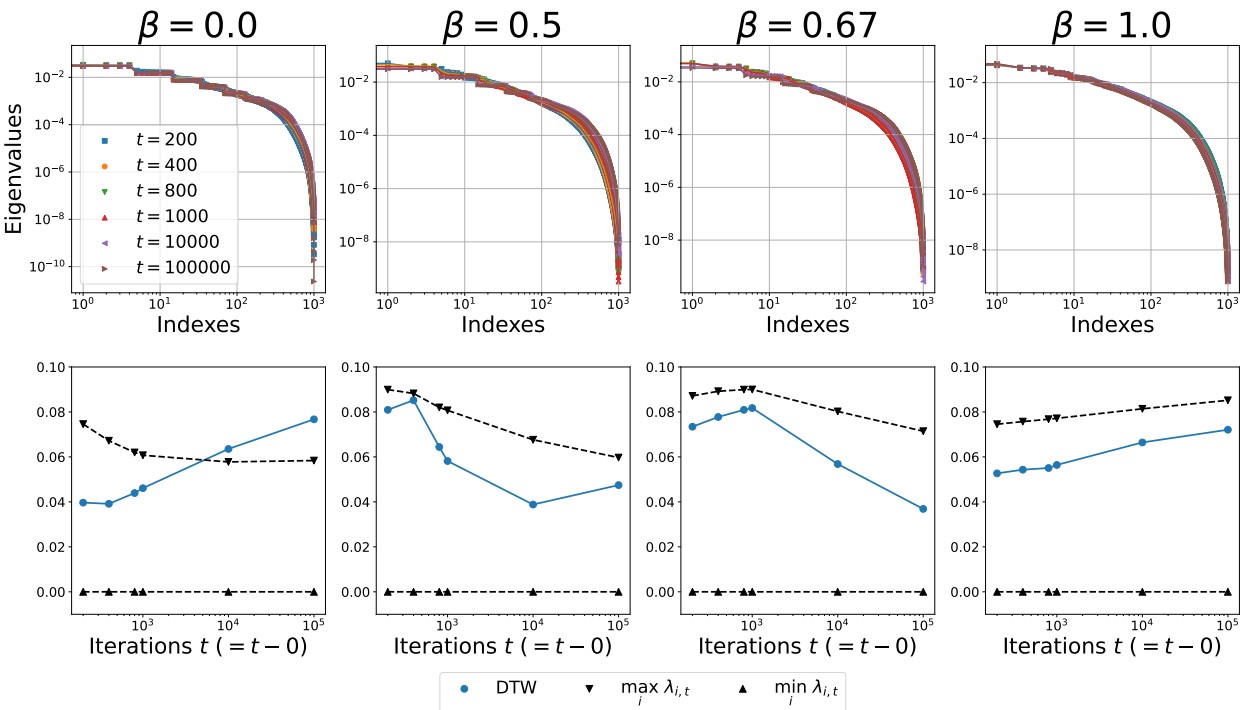

Figure 2: Behavior of eigenvalues $\boldsymbol{\lambda}_t = \{\lambda_{i,t}\}_{i=1}^m$ (first row) and the dynamic time warping (DTW) (second row) when we use the RBF kernel for SVGD with 1000 particles ($m = 1000$) in the Gaussian mixture experiments. We show the results up to the point where the KL divergence has not converged ($t = 200$ to $10^3$), as well as the results after convergence ($t = 10^4, 10^5$) according to Figure 6.

Before outlining the proof, we position our results in comparison to existing studies. As suggested by Korba et al. (2021) and Duncan et al. (2023), it is difficult for SVGD to achieve linear convergence in KL divergence and the difficulty also surfaces in our analysis. To show our results, the step size must be $\gamma_t \leq \mathcal{O}(1/t^{2/3})$ to control $\left\| \nabla \log \frac{\mu_t}{\pi} \right\|_{L^2(\mu_t)}$, which highlights the difficulty of achieving convergence faster than sub-linear order. While Huang et al. (2023) has shown the linear convergence in a continuous-time setting, the kernel function utilized in their study is specifically designed to guarantee linear convergence and thus it is not commonly employed in practice. On the other hand, our result is established within the discrete time setting that corresponds to the SVGD algorithm, under realistic assumptions commonly met by the RBF kernel frequently adopted in SVGD.

Expanding our sight to other deterministic sampling methods based on kernel functions, sub-linear convergence has been demonstrated in the kernel herding (e.g., Chen et al. (2010); Bach et al. (2012)) and Bayesian Quadrature context (e.g., Briol et al. (2015); Futami et al. (2019)) when employing infinite-dimensional kernel functions like the RBF kernel. Our results are consistent with these facts.

### 4.2 Spectral decomposition and $(\epsilon, \delta)$-approximation

The main objective here is to provide an overview of the proof focusing on how we detect $\epsilon_t$ and $\delta_t$ in the AGF. The complete proof is in Appendix C.

To conduct analyses based on our AGF, we need to show that $\left\| \nabla \log \frac{\mu_t}{\pi} \right\|_{L^2(\mu_t)}$ is bounded for all $t$ in SVGD, which is guaranteed by the following lemma (see Appendix C.2 for complete proof).

**Lemma 2.** *Suppose that Assumptions 1-3 and 5 are satisfied. Let $\gamma_t$ satisfies $\gamma_t \leq C_\gamma$ defined in Theorem 1. Then, there exists a positive problem-dependent constant $c$ and is independent of $t$ such that, for any $t \in (0, T]$ we have $\left\| \nabla \log \frac{\mu_t}{\pi} \right\|_{L^2(\mu_t)} \leq \left\| \nabla \log \frac{\mu_0}{\pi} \right\|_{L^2(\mu_0)} + c \sum_{k=0}^{t-1} \gamma_k$.*

Now we are ready to begin the analysis of the convergence of SVGD based on AGF. Substituting $g_{\mu_t}(x) = P_{\mu_t,k}\nabla\log\frac{\mu_t}{\pi}$ into Eq. (9) and multiplying both sides by $\eta_t(>0)$ yields

$$-\eta_t\left\langle\nabla\log\frac{\mu_t}{\pi},P_{\mu_t,k}\nabla\log\frac{\mu_t}{\pi}\right\rangle_{L^2(\mu_t)} = -\eta_t I_{\text{stein}}(\mu_t|\pi) \leq -\epsilon_t\eta_t\left\|\nabla\log\frac{\mu_t}{\pi}\right\|^2_{L^2(\mu_t)} + \eta_t\delta_t. \tag{13}$$

According to the fact that $\eta_t I_{\text{stein}}(\mu_t|\pi) \leq \text{KL}(\mu_0|\pi)$ (see Appendix C), we further obtain the following inequalities:

$$\text{KL}(\mu_0|\pi) \geq \eta_t I_{\text{stein}}(\mu_t|\pi) \geq \epsilon_t\eta_t\left\|\nabla\log\frac{\mu_t}{\pi}\right\|^2_{L^2(\mu_t)} - \eta_t\delta_t. \tag{14}$$

Therefore, our goal is to guarantee the existence of the above inequality. If Eq. (14) exists, we can qualitatively analyze the convergence in KL divergence by specifying $\{\epsilon_t,\delta_t\}$ and utilizing the property of AGF shown in Lemma 1 and Remarks 1 and 2.

To focus on the discussion for detecting $\{\epsilon,\delta\}$, we first mention the necessary conditions for the existence of Eq. (14) with respect to $\eta_t$ under our final results. As can be seen from Theorem 1, we obtain $\epsilon_t = c_0$ and $\delta_t = 0$ through the proof that we explain later, where $c_0$ is independent of $t$. In this case, from Eq. (13), it is necessary for $\eta_t\left\|\nabla\log\frac{\mu_t}{\pi}\right\|^2_{L^2(\mu_t)}$ to be uniformly upper bounded with respect to $t$ to compensate for the convergence based on AGF. This condition can be satisfied by setting $\eta_t$ such that it fulfills $\gamma_t \leq \mathcal{O}(1/t^{2/3})$ from Lemma 2 (see Appendix C for this derivation).

Our strategy is to show the boundedness of the following equality expressed as

$$\epsilon_t\eta_t\left\|\nabla\log\frac{\mu_t}{\pi}\right\|^2_{L^2(\mu_t)} - \eta_t I_{\text{stein}}(\mu_t|\pi) = \eta_t\left\langle\nabla\log\frac{\mu_t}{\pi},(\epsilon_t I - P_{\mu_t,k})\nabla\log\frac{\mu_t}{\pi}\right\rangle_{L^2(\mu_t)}. \tag{15}$$

We adopt the spectral decomposition of the kernel operator to analyze the above. Since a Hilbert–Schmidt operator $P_{\mu,k}$ is compact and self-adjoint, we have, for all $i$, $P_{\mu,k}\phi_i = \lambda_i\phi_i$, where $\phi_i \in L^2(\mu)$ represents an eigenfunction that satisfies a complete orthonormal system (CONS), and $\lambda_i$ is an eigenvalue corresponding to $\phi_i$. Even if these eigenvalues are ordered as $\lambda_1 \geq \lambda_2 \geq \ldots > 0$, it does not compromise generality. Moreover, the kernel function can be decomposed into $k(x,y) = \sum_{i=1}^\infty \lambda_i\phi_i(x)\phi_i(y)$, where the convergence of this infinite series holds in the norm of $\|\cdot\|_{L^2(\mu)}$.

Defining $v_t := \nabla\log\frac{\mu_t}{\pi}$ for simplicity in notation, we can obtain $v_t = \sum_{i=1}^\infty\langle v_t,\phi_i\rangle_{L^2(\mu)}\phi_i$ and $P_{\mu,k}v_t = \sum_{i=1}^\infty\lambda_i\langle v_t,\phi_i\rangle_{L^2(\mu)}\phi_i$ because the kernel function is dense in $L^2(\mu)$ and thus its eigenvectors are complete. We provide the discussion for non-complete eigenvectors in Appendix A. Substituting these equalities into Eq. (15), we have

$$\epsilon_t\eta_t\left\|\nabla\log\frac{\mu_t}{\pi}\right\|^2_{L^2(\mu_t)} - \eta_t I_{\text{stein}}(\mu_t|\pi) = \eta_t\sum_{i=1}^\infty(\epsilon_t - \lambda_i)\langle v_t,\phi_i\rangle^2_{L^2(\mu_t)}. \tag{16}$$

In the right-hand side term of the above, there exists a index $1 < j$ such that $\lambda_j > \epsilon_t > \lambda_{j+1}$ by setting sufficiently small $\epsilon_t$. Hence, by regularizing $\{\langle v_t,\phi_i\rangle^2_{L^2(\mu_t)}\}_{i=1}^\infty$, we can render the left-hand side of Eq. (16) negative. For that purpose, we focus on the RKHS associated with $k$ given as $\mathcal{H} = \{f \in L^2(\mu) \mid f = \sum_{i=1}^\infty a_i\phi_i, \sum_{i=1}^\infty\lambda_i^{-1}\|a_i\|^2 < \infty, a_i \in \mathbb{R}\}$, where $\mathcal{H}$ is dense in $L^2(\mu)$. In this RKHS, there exists a function $v_t^{(l)} \in \mathcal{H}$ such that the sequence of $v_t^{(l)} \to v_t$ as $l \to \infty$ in $L^2(\mu)$ norm. Thus, by approximating the original $v_t$ with $v_t^{(l)}$ in $\mathcal{H}$, we can regularize $\{\langle v_t,\phi_i\rangle^2_{L^2(\mu_t)}\}_{i=1}^\infty$.

Under the regularized $\{\langle v_t,\phi_i\rangle^2_{L^2(\mu_t)}\}_{i=1}^\infty$ in the above and sufficiently small $\epsilon_t$, we can obtain $\epsilon_t\eta_t\left\|\nabla\log\frac{\mu_t}{\pi}\right\|^2_{L^2(\mu_t)} - \eta_t I_{\text{stein}}(\mu_t|\pi) < 0$, which implies that $\delta_t = 0$ in the AGF. From Assumption 6, we can show that $\epsilon_t$ is the constant order with respect to $t$ and express it as $c_0$ (see Appendix C). This concludes the proof outline.

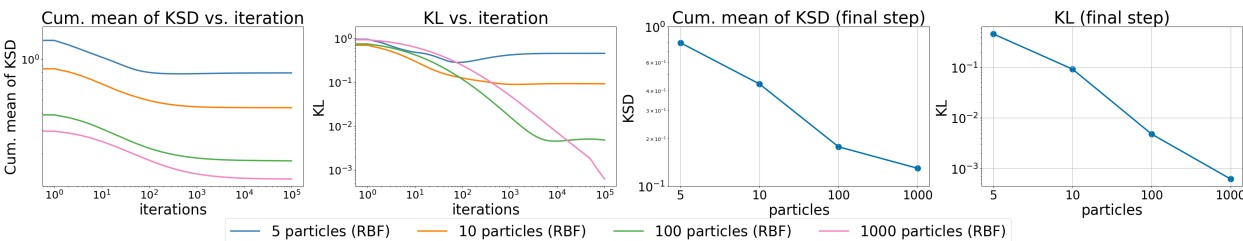

Figure 3: The convergence behavior in terms of $\mathrm{KL}(\mu_T|\pi)$ and $\frac{1}{T}\sum_{t=1}^{T} I_{\mathrm{stein}}(\mu_t|\pi)$ for all $T$ under two-dimensional Gaussian distribution experiments ($\beta = 0.67 \approx 2/3$).

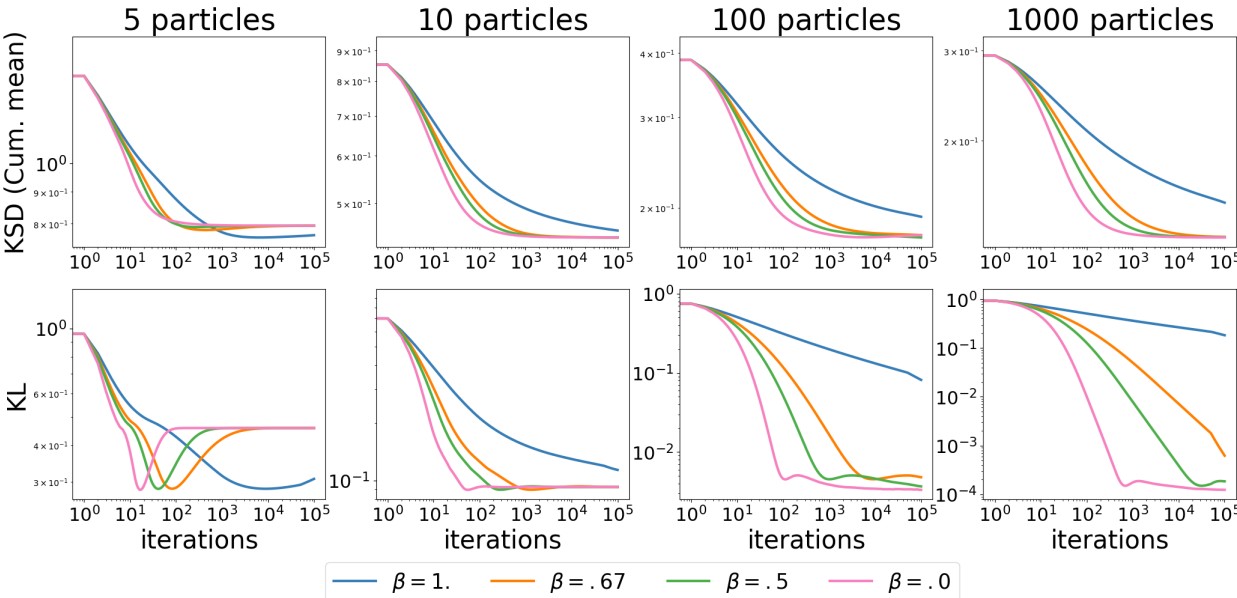

Figure 4: Convergence in $\mathrm{KL}(\mu_T|\pi)$ and $\frac{1}{T}\sum_{t=1}^{T} I_{\mathrm{stein}}(\mu_t|\pi)$ for all $T$ under different particles and stepsize settings ($\beta = \{0., 0.5, 0.67, 1.\}$).

## 5 Numerical experiments

In this section, we aim to confirm the validity of our theoretical results. We only show the results of the two-dimensional Gaussian experiments due to the page limitation. The details of the experimental settings and additional results including the Gaussian mixture can be seen in Appendix E.

We set the target distribution as the two-dimensional Gaussian distribution. We adopted the RBF kernel $k(x, y) = \exp(\frac{1}{h}\|x - x'\|_2^2)$, which is commonly used in practice and satisfies the assumptions in Section 4. The bandwidth $h$ was selected by the median trick as in Liu & Wang (2016). To appropriately verify our theoretical analysis, we simply set the decaying step size $\gamma_t = 1/(1 + t^\beta)(= \mathcal{O}(1/t^\beta))$ suggested by Theorem 1 and did not use the Adagrad-based stepsize, which is adopted in related studies such as Korba et al. (2021) and others. We evaluated the KL divergence: $\mathrm{KL}(\mu_T|\pi)$ and the cumulative mean of KSD: $\frac{1}{T}\sum_{t=1}^{T} I_{\mathrm{stein}}(\mu_t|\pi)$, which are theoretically guaranteed sub-linear convergence.

**Results:** From Figures 3 and 4, we can see that SVGD with the RBF kernel tends to achieve sub-linear convergence both in $\mathrm{KL}(\mu_T|\pi)$ and in $\frac{1}{T}\sum_{t=1}^{T} I_{\mathrm{stein}}(\mu_t|\pi)$, which supports Theorem 1. These results also offer two key takeaways for practitioners.

First, the empirical support for our Theorem 1 gives practitioners greater confidence that SVGD is indeed converging towards the target distribution in terms of KL divergence. Furthermore, our use of a theoretically-grounded step-size schedule, $\gamma_t = 1/(1 + t^\beta)$, offers a principled alternative to purely heuristic choices, which is a common challenge in practice.

Second, our experiments reveal a critical and non-intuitive trade-off between the number of particles and convergence dynamics. As expected, increasing the number of particles reduces the final bias in the KL divergence, since the approximation error $\delta_t$ in our AGF framework becomes smaller. However, we observe that employing a substantial number of particles leads to slower convergence for both KSD and KL divergence. This phenomenon is attributed to the properties of the RBF kernel; with more particles, the kernel operator $P_{\mu,k}$ has more exceedingly small eigenvalues, which correspond to directions in the function space where the gradient signal is weak, thus hindering the optimization process (see Appendix E for details). This presents a crucial trade-off for practitioners: for a given computational budget, one must choose between faster convergence to a slightly more biased solution (using fewer particles) and slower convergence to a more accurate, lower-bias solution (using more particles). This choice directly depends on the specific application's requirements for accuracy versus computational efficiency.

## 6 Limitation & Conclusion

Ensuring the convergence of SVGD in KL divergence has proven challenging in finite and infinite particle settings. Furthermore, while many studies have provided convergence guarantees for SVGD in KSD, these do not necessarily ensure its weak convergence. As a *first strategy to address this issue*, we conducted the convergence analysis of SVGD under the ideal conditions of an infinite particle setting that guarantees an accurate gradient approximation. Then, we successfully elucidated the convergence of SVGD in KL divergence in this setting. Our finding suggests weak convergence of SVGD with infinite particles, affirming its capability to approximate the expectation by the target distribution without bias, akin to MCMC.

One of limitations in our paper is the challenge in furnishing a theoretical explanation for the convergence of SVGD when employing a finite number of particles. Extending our analysis to finite particle settings using AGF is being considered as our future study. The main challenge in this extension is expected to be in determining the values of $\epsilon$ and $\delta$, primarily due to the unknown theoretical properties of gradient approximation on RKHS when dealing with correlated particles, as far as our current knowledge extends. Given these theoretical hurdles, an empirical characterization of the bias term $\delta_t$ for finite, correlated particles would be a valuable, albeit challenging, component of this future work.

Another limitation is that Assumption 6 is rather strong. This assumption, introduced to ensure that $c_0$ remains of constant order with respect to $t$, is difficult to justify in the infinite particle setting. The pursuit of convergence guarantees grounded in milder assumptions represents a crucial avenue for future research. Furthermore, we aspire for this study to catalyze further research endeavors that aim to furnish better convergence guarantee for SVGD in KL divergence.

## Acknowledgment

We thank anonymous reviewers for their valuable feedback and discussions. MF was previously supported by the RIKEN Special Postdoctoral Researcher Program and JST ACT-X Grant Number JPMJAX210K, Japan. MF is currently supported by KAKENHI Grant Number 25K21286. FF was supported by JSPS KAKENHI Grant Number JP23K16948. FF was supported by JST, PRESTO Grant Number JPMJPR22C8, Japan.

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

# A Discussion regarding the assumptions

It should be mentioned that Assumptions 3-5 is satisfied in the RBF kernel commonly used in SVGD. Assumption 3 requires the bounded twice differentiability of the kernel, which is stronger than existing work (Salim et al., 2022) but is required to control $\|\nabla \log \frac{\mu_t}{\pi}\|_{L^2(\mu_t)}$ in Lemma 2. The existence of the constant $B$ in this assumption depends on the choice of bandwidth for the RBF kernel and can be guaranteed by adopting standard selection methods such as the median trick (Liu & Wang, 2016). Assumption 5 holds true since the RBF kernel satisfies $\int k(x,x)\mathrm{d}\mu_t(x) = 1$. Assumption 6 is a crucial technical condition required to prove Theorem 1. As outlined in Section 4.2 and detailed in the full proof in Appendix C, our analysis aims to show that SVGD is a $(c_0, 0)$-AGF, where $c_0$ is a constant independent of the iteration time $t$. This constant $c_0$ is derived from the eigenvalues $\lambda_i$ of the kernel operator. If the eigenvalues were allowed to decay to zero as $t \to \infty$, then the relative approximation quality $\epsilon_t$ (which we show is a constant $c_0$) would also decay. This would prevent us from applying the LSI to obtain the final sub-linear convergence rate. Thus, the assumption of time-invariant positive bounds on the eigenvalues is what allows us to establish a time-independent $\epsilon_t = c_0 > 0$.

While formally proving this assumption in the infinite-particle setting is challenging, we, at the very least, provide empirical justification in Appendix E. As our numerical experiments show (see Figures Figures 9-12), the eigenvalues tend to become independent of time as the number of particles increases, supporting the use of this assumption as a plausible idealization at least in the Gaussian and Gaussian mixture distribution estimation experiments. The need for better proof remains an important future task.

Assumption 4 assures that the projection $P_{\mu,k}$ is injective and characteristic (Sriperumbudur et al., 2010), and also implies certain types of universality (Sriperumbudur et al., 2011), which hold in the RBF kernel. Thanks to these properties, we can expand the Wasserstein gradient via the eigenvectors as Eq. (16) since the eigenvectors of $P_{\mu,k}$ can be CONS in $L^2(\mu)$. Furthermore, SVGD ensures $\delta_t = 0$ under Assumption 4, a critical condition for convergence in KL divergence (see Section 3.2), which sets it apart from other approximate flow methods that face challenges in achieving $\delta_t = 0$. From these discussions, it is apparent that Assumption 4 plays a pivotal role to guarantee sub-linear convergence of SVGD in KL divergence, while being a stronger assumption than in existing studies (Korba et al., 2020; Salim et al., 2022; Sun et al., 2023).

If the eigenvectors of $P_{\mu,k}$ is not CONS in $L^2(\mu)$, there is a bias to approximate $v_t := \nabla \log \frac{\mu_t}{\pi}$ in RKHS. For instance, when we consider the null space of $P_{\mu,k}$ as $\mathrm{Null}(P_{\mu,k}) := \{f \in L^2(\mu) | P_{\mu,k}f = 0\}$, we have $v_t = \sum_{i=1}^{\infty} \langle v_t, \phi_i \rangle_{L^2(\mu)} \phi_i + \Psi$, where $\Psi \in \mathrm{Null}(P_{\mu,k})$ is some appropriately chosen function. In this case, we can express the right-hand side of Eq. (15) as $\eta_t \sum_{i=1}^{\infty} (\epsilon_t - \lambda_i) \langle v_t, \phi_i \rangle_{L^2(\mu)}^2 + \eta_t \|\Psi\|_{L^2(\mu)}^2$, which implies $\delta_t = \|\Psi\|_{L^2(\mu)}^2 (\neq 0)$ in the AGF. Thus, from Remark 2, the KL divergence does not go to 0 even when we increase $T$. Such cases may arise when using linear or polynomial kernels in SVGD since these do not satisfy Assumption 4. Also, the eigenvectors are not guaranteed to be CONS when using a finite number of particles. If we draw $m$ independent and identically distributed (i.i.d.) samples from $\mu(x)$ and approximate the kernel operator, we can only access the first $m$-dimensional eigenvectors (Rosasco et al., 2010) and thus corresponding eigenvectors cannot be CONS in $L^2(\mu)$. Note that when considering the SVGD with finite particles, since the particles are not i.i.d. and correlated significantly, the approximation quality is much worse than the case of i.i.d. particles and results in larger $\delta_t$.

# B Preliminaries

## B.1 Gradient flow

We are interested in sampling from the density $\pi \in \mathcal{P}_2(\mathcal{X})$ and proportional as $\pi \propto \exp^{-V(x)}$, where $V : \mathcal{X} \to \mathbb{R}$. We assume that $V \in C^2(\mathcal{X})$ and its Hessian $H_V$ satisfies $\|H_V\|_{\mathrm{op}} \leq L$.

We are interested in sampling from $\pi$ and formalizing the task as the optimization problem. For that purpose, We define the Kullback-Leibler (KL) divergence. For any $\mu, \pi \in \mathcal{P}_2(\mathcal{X})$, KL divergence of $\mu$ w.r.t $\pi$ is defined

by

$$\mathrm{KL}(\mu|\pi) := \int \log \frac{\mathrm{d}\mu}{\mathrm{d}\pi}(x)\mathrm{d}\mu(x), \tag{17}$$

if $\mu$ is absolutely continuous w.r.t $\pi$ and admits Radon-Nikodym derivative $\mathrm{d}\mu/\mathrm{d}\pi$ and $\mathrm{KL}(\mu|\pi) = +\infty$ otherwise. We consider a functional $\mathrm{KL}(\cdot|\pi) : \mathcal{P}_2(\mathcal{X}) \to [0, +\infty)$, $\mu \to \mathrm{KL}(\mu|\pi)$ defined on over Wasserstein space:

$$\min_{\mu \in \mathcal{P}_2(\mathcal{X})} \mathcal{F}(\mu), \quad \mathcal{F}(\mu) := \mathrm{KL}(\mu|\pi). \tag{18}$$

The advantage of using Wasserstein space is its differential structure to minimize this kind of functional.

Before introducing the differential structure, we introduce the continuous map. Let $T > 0$ and consider a weakly continuous map $\mu : (0, T) \to \mathcal{P}_2(\mathcal{X})$, $t \to \mu_t$. The family $(\mu_t)_{t \in (0,T)}$ satisfies a continuity equation if there exists $(v_t)_{t \in (0,T)}$ such that $v_t \in L^2(\mu_t)$:

$$\frac{\partial \mu_t}{\partial t} + \mathrm{div}(\mu_t v_t) = 0, \tag{19}$$

holds in the sense of distributions, i.e., for any $\phi \in C_c^\infty(\mathcal{X})$,

$$\frac{\mathrm{d}}{\mathrm{d}t} \int \phi(x)\mathrm{d}\mu_t(x) = \langle \nabla\phi, v_t \rangle_{L^2(\mu_t)} = \int \langle \nabla\phi(x), v_t(x) \rangle \mathrm{d}\mu_t(x), \tag{20}$$

holds for any $t \in (0, T)$. And by integration parts, we have

$$\int \phi(x)\frac{\partial \mu_t(x)}{\partial t} + \int \phi(x)\mathrm{div}(\mu_t(x)v_t(x)) = 0. \tag{21}$$

Although the Wasserstein space is not a Riemannian manifold, it can be equipped with a Riemannian structure and interpretation (Otto, 2001) and the tangent space of $\mathcal{P}_2(\mathcal{X})$ at $\mu_t$ denoted $\mathcal{T}_{\mu_t}\mathcal{P}_2(\mathcal{X})$ is a subset of $L^2(\mu_t)$. Under this setting, among all possible $(v_t)_{t \in (0,T)}$, we call $v_t$ that has the minimal $L^2(\mu_t)$ norm as the velocity field of $(\mu_t)_{t \in (0,T)}$ and this minimality condition can be characterized by $v_t \in \mathcal{T}_{\mu_t}\mathcal{P}_2(\mathcal{X}) \subset L^2(\mu_t)$.

To select the appropriate $(v_t)_{t \in (0,T)}$, it is useful to use the differential structure of the Wasserstein space. Assume that given a proper lower semi-continuous functional $\mathcal{F} : \mathcal{P}_2(\mathcal{X}) \to \mathbb{R}$ and $\mu \in \mathcal{P}_2(\mathcal{X})$, $\xi \in L^2(\mu)$ is a strong sub-differential of $\mathcal{F}$ at $\mu$ if for every $\phi \in L^2(\mu)$ and for every $\epsilon \in (0, 1]$,

$$\mathcal{F}(\mu) + \epsilon \langle \xi, \phi \rangle_{L^2(\mu)} + o(\epsilon) \le \mathcal{F}((I + \epsilon\phi)\#\mu). \tag{22}$$

where $I$ is the identity map.

Then the important consequence is that under the mild regularity conditions, $W_2$ gradient of $\mathcal{F}$ corresponds to the first variation of the functional. Assume that given a functional $\mathcal{F} : \mathcal{P}_2(\mathcal{X}) \to \mathbb{R}$, we call $\frac{\delta\mathcal{F}}{\delta\mu}(\mu)$ as the first variation of $\mathcal{F}$ at $\mu$

$$\int \frac{\delta\mathcal{F}}{\delta\mu}(\mu)\phi d\xi = \frac{\mathrm{d}}{\mathrm{d}\epsilon}\mathcal{F}(\mu + \epsilon\xi)\Big|_{\epsilon=0} = \lim_{\epsilon \to 0} \frac{\mathcal{F}(\mu + \epsilon\xi) - \mathcal{F}(\mu)}{\epsilon}, \tag{23}$$

for all $\xi = \nu - \mu$ where $\nu \in \mathcal{P}_2(\mathcal{X})$.

From Lemma 10.4.1 in Ambrosio et al. (2005), given $\mu \in \mathcal{P}_2(\mathcal{X})$, which is absolutely continuous with respect to the Lebesgue measure and its density is $C^1(\mathcal{X})$ and assume that $\nabla_{W_2}\mathcal{F}(\mu)(x)$ belongs to the strong sub-differential of $\mathcal{F}$ at $\mu$. Then it is given as

$$\nabla_{W_2}\mathcal{F}(\mu)(x) = \nabla\frac{\delta\mathcal{F}}{\delta\mu}(\mu)(x) \quad \text{for } \mu - \text{a.e. } x \in \mathbb{R}^d, \tag{24}$$

and, for every vector field $\xi \in C_c^\infty(\mathbb{R}^d, \mathbb{R}^d)$,

$$\langle \nabla_{W_2} \mathcal{F}(\mu), \xi \rangle_{L^2(\mu)} = -\int_{\mathbb{R}^d} \nabla \frac{\delta \mathcal{F}}{\delta \mu}(\mu)(x) \mathrm{div}(\mu(x)\xi(x)) dx, \tag{25}$$

where $\nabla_{W_2} \mathcal{F}(\mu)$ belongs to $\mathcal{T}_\mu \mathcal{P}(\mathbb{R}^d)$, which is a subset of $L^2(\mu)$.

Now we are ready to leverage this differential structure of the Wasserstein space to minimize the functional $\mathcal{F}$. We consider how $\mathcal{F}$ evolves by the continuity equation.

$$\dot{\mathcal{F}}(\mu_t) := \frac{\mathrm{d}}{\mathrm{d}t} \mathcal{F}(\mu_t) = \langle \nabla_{W_2} \mathcal{F}(\mu_t), v_t \rangle_{L^2(\mu_t)}. \tag{26}$$

Thus, by choosing $v_t$ such that $\langle \nabla_{W_2} \mathcal{F}(\mu_t), v_t \rangle_{L^2(\mu_t)} \leq 0$, we can minimize the functional by using the continuity equation. A natural choice is to use the Wasserstein gradient itself as $v_t = -\nabla_{W_2} \mathcal{F}(\mu_t)$, which results in

$$\dot{\mathcal{F}}(\mu_t) := \frac{\mathrm{d}}{\mathrm{d}t} \mathcal{F}(\mu_t) = -\|\nabla_{W_2} \mathcal{F}(\mu_t)\|_{L^2(\mu_t)}^2 \leq 0. \tag{27}$$

## B.2 KL divergence

Now we focus on minimizing KL divergence. It is known that the Wasserstein gradient of the KL divergence is given as $\nabla_{W_2} \mathrm{KL}(\mu|\pi) = \nabla \log \frac{\mu}{\pi} \in L^2(\mu)$. Then by setting $v_t = -\nabla_{W_2} \mathrm{KL}(\mu|\pi) = -\nabla \log \frac{\mu}{\pi}$ as the velocity field of the continuity equation, we have that

$$\frac{\mathrm{d}}{\mathrm{d}t} \mathrm{KL}(\mu_t|\pi) = -\left\|\nabla \log \frac{\mu}{\pi}\right\|_{L^2(\mu_t)}^2. \tag{28}$$

When considering the forward discretization, we obtain the gradient descent algorithm in the Wasserstein space

$$\mu_{t+1} = \left(I - \gamma \nabla \log \frac{\mu_n}{\pi}\right) \#\mu_t, \tag{29}$$

where $\gamma > 0$ is a stepsize.

To analyze this time evolution in the continuous dynamics, many existing work assumes that $\pi$ satisfies the logarithmic Sobolev inequality (LSI) (Bakry et al., 2013). The important consequence of this inequality is that there exists a positive constant $C_{\mathrm{LS}}$ such that

$$\mathrm{KL}(\mu_t|\pi) \leq \frac{1}{C_{\mathrm{LS}}} \left\|\nabla \log \frac{\mu}{\pi}\right\|_{L^2(\mu_t)}^2. \tag{30}$$

With this inequality, we have that

$$\mathrm{KL}(\mu_t|\pi) \leq e^{-C_{\mathrm{LS}}t} \mathrm{KL}(\mu_0|\pi). \tag{31}$$

Thus, the Wasserstein gradient flow of the KL divergence achieves linear convergence.

How can we implement $\nabla_{W_2} \mathrm{KL}(\mu|\pi) = \nabla \log \frac{\mu}{\pi} \in L^2(\mu)$ in practice ? It is known that the Langevin dynamics are given as

$$\mathrm{d}X_t = \nabla \log \pi(X_t) \mathrm{d}t + \sqrt{2} \mathrm{d}B_t, \tag{32}$$

where $\mathrm{d}B_t$ is the standard Brownian motion in $\mathbb{R}^d$ can be seen as the implementation of the Wasserstein gradient flow in a probabilistic way. We can easily confirm that the probability density induced by SDE of Eq. 32 is given as the Fokker–Planck(FP) equation and that of the FP equation is equivalent to the continuity equation given by the Wasserstein gradient flow in a distributional sense Jordan et al. (1998).

### B.3 Stein variational gradient descent (SVGD)

The LD method was successfully used as an MCMC method, however, it suffers from large bias when using obtained samples. To alleviate this, an alternative gradient flow approach has been developed. Among those methods, SVGD has been used extensively in practice. Since SVGD is a kind of projection of the Wasserstein gradient into the reproducing kernel Hilbert space (RKHS), here we first introduce the settings of RKHS.

Let a semi-positive definite kernel $k : \mathcal{X} \times \mathcal{X} \to \mathbb{R}$ and $\mathcal{H}_0$ is its corresponding RKHS of real-valued functions $\mathcal{X} \to \mathcal{R}$. We express the inner product $\langle \cdot, \cdot \rangle_{\mathcal{H}_0}$. Due to the reproducing property $\forall f \in \mathcal{H}_0$, $f = \langle f, k(x, \cdot) \rangle_{\mathcal{H}_0}$. We define by $\mathcal{H}$ as the Cartesian product of $\mathcal{H}_0$, its element $f \in \mathcal{H}$, $f = (f_1, \ldots, f_d)$ and $f_i \in \mathcal{H}_0$ for $i = 1, \ldots, d$. Then the inner product of $\mathcal{H}$ is given as $\langle f, g \rangle_{\mathcal{H}} = \sum_{i=1}^{d} \langle f_i, g_i \rangle_{\mathcal{H}_0}$. Let $\mu \in \mathcal{P}_2(\mathcal{X})$ and if $\int k(x, x) \mathrm{d}\mu(x) < \infty$, then the integral operator associated to $k$ and $\mu$ denoted by $S_{\mu,k} : L^2(\mu) \to \mathcal{H}$ is defined as

$$S_{\mu,k} f := \int k(x, \cdot) f(x) \mathrm{d}\mu(x). \tag{33}$$

By definition, note that $\mathcal{H} \subset L^2(\mu)$. We also define the inclusion map as $\iota : \mathcal{H} \to L^2(\mu)$ and it is the adjoint of $S_{\mu,k}$. Thus for $f \in L^2(\mu)$ and $g \in \mathcal{H}$, we have

$$\langle f, \iota g \rangle_{L^2(\mu)} = \langle \iota^* f, g \rangle_{\mathcal{H}} = \langle S_{\mu,k} f, g \rangle_{\mathcal{H}}. \tag{34}$$

We also define $P_{\mu,k} : L^2(\mu) \to L^2(\mu)$, $P_{\mu,k} = \iota S_{\mu,k}$.

In SVGD, instead of using the Wasserstein gradient directly, we consider using $P_{\mu,k} \nabla \log \frac{\mu}{\pi}$ as $v_t$. Then we obtain the discretized dynamics of SVGD

$$\mu_{t+1} = \left( I - \gamma P_{\mu,k} \nabla \log \frac{\mu_n}{\pi} \right) \#\mu_t, \tag{35}$$

and if the kernel satisfies $\lim_{\|x\| \to \infty} k(x, \cdot) \pi(x) = 0$, we obtain

$$P_{\mu,k} \nabla \log \frac{\mu}{\pi}(\cdot) := -\int [\nabla \log \pi(x) k(x, \cdot) + \nabla_x k(x, \cdot)] \mathrm{d}\mu(x), \tag{36}$$

by using an integration by parts (Liu, 2017). Then, approximating the expectation of $\mu$ as in Liu & Wang (2016), preparing initial samples $\{x_0^m\}_{m=1}^{M}$ and iteratively updating them by a transformation, resulting in

$$x_{t+1}^m = x_n^m - \frac{1}{M} \sum_{m'=1}^{M} \nabla \log \pi(x_n^{m'}) k(x_n^{m'}, x_n^m) + \nabla_x k(x_n^{m'}, x_n^m), \tag{37}$$

where we approximate $\mu_n$ by a finite set of particles $\hat{\mu}_n = \frac{1}{M} \sum_{m=1}^{M} \delta_{x_n^m}$ where $\delta_x$ is the Dirac measure with its mass at $x$. Originally, Liu & Wang (2016) derived the push forward as follows. Assume that the pushforward is given as $T(x) = x - \gamma \phi(x)$, where $\gamma$ is a positive constant $\phi(\cdot) \in \mathcal{H}$ is a perturbation direction. Then the update direction, which maximally decreases the Kullback–Leibler (KL) divergence between the particles and the target distribution,

$$\phi^*(x) = \underset{\phi \in \mathcal{H}, \|\phi\|_{\mathcal{H}} \leq 1}{\arg\max} \left\{ -\frac{\mathrm{d}}{\mathrm{d}\gamma} \mathrm{KL}(T \#\mu | \pi)|_{\gamma=0} \right\}, \tag{38}$$

then the solution of this is $\phi^*(\cdot) = S_{\mu,k} \nabla \log \frac{\mu}{\pi}$.

Going back to the continuous dynamics, we have

$$\begin{aligned}
\frac{\mathrm{d}}{\mathrm{d}t} \mathrm{KL}(\mu_t | \pi) &= -\left\langle \nabla \log \frac{\mu_t}{\pi}, P_{\mu_t,k} \nabla \log \frac{\mu_t}{\pi} \right\rangle_{L^2(\mu_t)} \\
&= -\left\langle \iota^* \nabla \log \frac{\mu_t}{\pi}, S_{\mu_t,k} \nabla \log \frac{\mu_t}{\pi} \right\rangle_{\mathcal{H}} \\
&= -\left\langle S_{\mu_t,k} \nabla \log \frac{\mu_t}{\pi}, S_{\mu_t,k} \nabla \log \frac{\mu_t}{\pi} \right\rangle_{\mathcal{H}} \\
&= -\left\| S_{\mu_t,k} \nabla \log \frac{\mu_t}{\pi} \right\|_{\mathcal{H}}^2 := -I_{\mathrm{stein}}(\mu | \pi),
\end{aligned} \tag{39}$$

where $I_{\text{stein}}(\mu|\pi)$ is called as the Stein–Fisher (SF) information (Duncan et al., 2023). It is known that the square root of the SF information corresponds to the KSD. Then it is natural to examine the inequality like LSI as follows:

$$\text{KL}(\mu|\pi) \leq cI_{\text{stein}}(\mu|\pi), \tag{40}$$

where $c$ is some positive constant and this is called Stein log-Sobolev inequality (Duncan et al., 2023). Unfortunately, the condition of this inequality is less clear compared to the LSI and Duncan et al. (2023) showed that it might fail to hold this inequality in many practical models like the RBF kernel function with exponential tail of $\pi$.

Thus it is difficult to consider the linear convergence of KL divergence under the geometry of $\mathcal{H}$. See Liu (2017) and Duncan et al. (2023) for a detailed discussion of the geometry of SVGD.

## C  Proofs of theories in Section 4

### C.1  Proof of Theorem 1

Under Assumptions 1 and 3-5, we have the following results from Theorem 3.2 in Salim et al. (2022):

$$\text{KL}(\mu_{t+1}|\pi) \leq \text{KL}(\mu_t|\pi) - \gamma_t \left(1 - \frac{\gamma_t B^2(\alpha^2 + L)}{2}\right) I_{\text{stein}}(\mu_t|\pi), \tag{41}$$

where $\eta_t := \gamma_t \left(1 - \frac{\gamma_t B^2(\alpha^2 + L)}{2}\right)$. Substituting $g_{\mu_t}(x) = P_{\mu_t,k} \nabla \log \frac{\mu_t}{\pi}$ into Eq. (9) yields

$$-\eta_t I_{\text{stein}}(\mu_t|\pi) \leq -\epsilon_t \eta_t \left\|\nabla \log \frac{\mu_t}{\pi}\right\|_{L^2(\mu_t)}^2 + \eta_t \delta_t. \tag{42}$$

To show that the above inequality exists and to evaluate $\epsilon_t$ and $\delta_t$, we study the following equality obtained via Eq. (42):

$$\epsilon_t \eta_t \left\|\nabla \log \frac{\mu_t}{\pi}\right\|_{L^2(\mu_t)}^2 - \eta_t I_{\text{stein}}(\mu_t|\pi) = \eta_t \left\langle \nabla \log \frac{\mu_t}{\pi}, (\epsilon_t I - P_{\mu_t,k}) \nabla \log \frac{\mu_t}{\pi}\right\rangle_{L^2(\mu_t)}. \tag{43}$$

As written in the main paper, we discuss the necessary conditions for the existence of Eq. (42) with respect to $\eta_t$ under our final results. As can be seen from Theorem 1, we obtain $\epsilon_t = c_0$ and $\delta_t = 0$ through the proof that we explain later, where $c_0$ is independent of $t$. In this case, as written in the main paper, it is necessary for $\eta_t \left\|\nabla \log \frac{\mu_t}{\pi}\right\|_{L^2(\mu_t)}^2$ to be uniformly upper bounded with respect to $t$ to compensate for the convergence based on AGF. To satisfy this condition, from Lemma 2, when we set $\gamma_t \leq \mathcal{O}(1/t^{2/3})$, the second term is the order of $\sum_t \gamma_t \leq \mathcal{O}(\eta^{1/3})$. Then $\left\|\nabla \log \frac{\mu_t}{\pi}\right\|_{L^2(\mu_t)}^2 \leq \mathcal{O}(\eta^{2/3})$. Then we can easily find that $\eta_t \left\|\nabla \log \frac{\mu_t}{\pi}\right\|_{L^2(\mu_t)}^2 \leq \mathcal{O}(1)$ with respect to $t$.

From here, we analyze Eq. (43) by using the spectral decomposition of the kernel operator. Since a Hilbert–Schmidt operator $P_{\mu,k}$ is compact and self-adjoint and we use the real-valued kernel function, we can decompose $P_{\mu,k}$ by spectral theorem. Thus, we have, for all $i$,

$$P_{\mu,k}\phi_i = \lambda_i \phi_i, \tag{44}$$

where $\phi_i \in L^2(\mu)$ represents an eigenfunction that satisfies a complete orthonormal system (CONS), and $\lambda_i$ is an eigenvalue corresponding to $\phi_i$. Even if these eigenvalues are ordered as $\lambda_1 \geq \lambda_2 \geq \ldots > 0$, it does not compromise generality. Moreover, the kernel function can be decomposed into $k(x, y) = \sum_{i=1}^{\infty} \lambda_i \phi_i(x) \phi_i(y)$, where the convergence of this infinite series holds in the norm of $\|\cdot\|_{L^2(\mu)}$.

From the spectral theorem, all the eigenvalues of a positive-definite kernel function are positive real values and their multiplicity (the dimension of the eigenspace) is finite. Under Assumption 4 (ISPD assumption), a

kernel function is dense in $L^2(\mu)$ (Steinwart & Christmann, 2008; Sriperumbudur et al., 2011; Carmeli et al., 2010). Therefore, the eigenfunctions $\{\phi_n\}$ are CONS in $L^2(\mu)$. Then, for any $f \in L^2(\mu)$, we have

$$f = \sum_{i=1}^{\infty} \langle f, \phi_i \rangle_{L^2(\mu)} \phi_i. \tag{45}$$

We should note that the above discussions overly simplify the eigenvalues and eigenvectors for vector functions because we treat the kernel function as the vector-valued one. As we mentioned in the main paper, since $f = (f_1, \cdots, f_d) \in L^2(\mu)$ is the vector valued function, each $f_1, \ldots, f_d \in L_0^2(\mu)$ are measurable function $f_1 : \mathcal{X} \to \mathbb{R}$ with $\int f_1^2 d\mu < \infty$. Abusing the notation, $P_{\mu,k} : L_0^2(\mu) \to \mathcal{H}_0$, which projects the scalar function to $\mathcal{H}_0$. In this case, the eigenvalues and eigenvectors are given as

$$P_{\mu,k}\psi_i = \lambda_i \psi_i, \tag{46}$$

where $\psi_i \in L_0^2(\mu)$ is an eigenfunction which is a scalar value function. When considering $P_{\mu,k} : L^2(\mu) \to \mathcal{H}$, the operator $P_{\mu,k}\phi$ is regarded as the elementwise projection defined as $P_{\mu,k}\phi_i = (\int k(x,y)\phi_1(y)d\mu(y), \cdots, \int k(x,y)\phi_d(y)d\mu(y))$. Since the eigenfunctions $\{\psi_i\}$ are CONS in $L_0^2(\mu)$ and the vector functions are in $L^2(\mu)$, we focus on each dimension and apply a spectral decomposition, that is,

$$f = (f_1, \ldots, f_d) = \left( \sum_{i=1}^{\infty} \langle f_1, \psi_i \rangle_{L_0^2(\mu)} \psi_i, \ldots, \sum_{i=1}^{\infty} \langle f_d, \psi_i \rangle_{L_0^2(\mu)} \psi_i \right). \tag{47}$$

By applying the kernel projection, we have

$$P_{\mu,k}f = (P_{\mu,k}f_1, \ldots, P_{\mu,k}f_d) = \left( \sum_{i=1}^{\infty} \lambda_i \langle f_1, \psi_i \rangle_{L_0^2(\mu)} \psi_i, \ldots, \sum_{i=1}^{\infty} \lambda_i \langle f_d, \psi_i \rangle_{L_0^2(\mu)} \psi_i \right). \tag{48}$$

The above equality corresponds to the setting when $\phi_i = (\psi_i, \cdots, \psi_i) \in L^2(\mu)$. Then, for any $f \in L^2(\mu)$, we have

$$f = \sum_{i=1}^{\infty} \langle f, \phi_i \rangle_{L^2(\mu)} \phi_i, \tag{49}$$

and

$$P_{\mu,k}f = \sum_{i=1}^{\infty} \lambda_i \langle f, \phi_i \rangle_{L^2(\mu)} \phi_i. \tag{50}$$

From the above equalities, we can be seen as the same calculation for $P_{\mu,t} : L_0^2(\mu) \to \mathcal{H}_0$ and $P_{\mu,t} : L^2(\mu) \to \mathcal{H}$.

In this paper, for simplicity, we do not work on $P_{\mu,t} : L_0^2(\mu) \to \mathcal{H}_0$ with eigenvectors $\{\psi_i\}$, but work on $P_{\mu,t} : L^2(\mu) \to \mathcal{H}$ with eigenvectors $\{\phi_i\}$ as shown in Eq. (44). For completeness, we remark the norm calculation as follows:

$$\|f\|_{L^2(\mu)}^2 = \sum_{j=1}^{d} \|f_j\|_{L_0^2(\mu)}^2 = \sum_{j=1}^{d} \sum_{i=1}^{\infty} \langle f_j, \psi_i \rangle_{L_0^2(\mu)}^2 = \sum_{i=1}^{\infty} \sum_{j=1}^{d} \langle f_j, \psi_i \rangle_{L_0^2(\mu)}^2, \tag{51}$$

where $\sum_{j=1}^{d} \langle f_j, \psi_i \rangle_{L_0^2(\mu)}^2$ is the multiply $\psi_i$ in an elementwise way and summing it up. Based on the above, we have

$$\|P_{\mu,k}f\|_{L^2(\mu)}^2 = \sum_{j=1}^{d} \|P_{\mu,k}f_j\|_{L_0^2(\mu)}^2 = \sum_{j=1}^{d} \sum_{i=1}^{\infty} \lambda_i \langle f_j, \psi_i \rangle_{L_0^2(\mu)}^2 = \sum_{i=1}^{\infty} \lambda_i \sum_{j=1}^{d} \langle f_j, \psi_i \rangle_{L_0^2(\mu)}^2. \tag{52}$$

By using the spectral decomposition and the Hilbert–Schmidt theorem, we obtain

$$k(x, y) = \sum_{i=1}^{\infty} \lambda_i \phi_i(x) \phi_i(y), \tag{53}$$

where the convergence of this infinite series holds in the norm of $\| \cdot \|_{L^2(\mu)}$. If $\mathcal{X}$ is a compact space, then from Mercer's theorem, the convergence of the above infinite series holds absolutely and uniformly. However, we *do not* assume that $\mathcal{X}$ is compact.

Let us show that, in our analysis, it is sufficient to study the convergence of $k(x, y) = \sum_{i=1}^{\infty} \lambda_i \phi_i(x) \phi_i(y)$ in $L^2(\mu)$ since our analysis is based on the norm of $\| \cdot \|_{L^2(\mu)}$. From Eq. (43), we have

$$\eta_t \left\langle \nabla \log \frac{\mu_t}{\pi}, (\epsilon_t I - P_{\mu_t, k}) \nabla \log \frac{\mu_t}{\pi} \right\rangle_{L^2(\mu_t)}$$

$$\leq \eta_t \left\langle \nabla \log \frac{\mu_t}{\pi}, \left( \epsilon_t I - \sum_{i=1}^{\infty} \lambda_i \phi_i(x) \phi_i(\cdot) + \sum_{i=1}^{\infty} \lambda_i \phi_i(x) \phi_i(\cdot) - P_{\mu_t, k} \right) \nabla \log \frac{\mu_t}{\pi} \right\rangle_{L^2(\mu_t)}$$

$$\leq \eta_t \left\langle \nabla \log \frac{\mu_t}{\pi}, \left( \epsilon_t I - \sum_{i=1}^{\infty} \lambda_i \phi_i(x) \phi_i(\cdot) \right) \nabla \log \frac{\mu_t}{\pi} \right\rangle_{L^2(\mu_t)}$$

$$+ \eta_t \left\langle \nabla \log \frac{\mu_t}{\pi}, \left( \sum_{i=1}^{\infty} \lambda_i \phi_i(x) \phi_i(\cdot) - P_{\mu_t, k} \right) \nabla \log \frac{\mu_t}{\pi} \right\rangle_{L^2(\mu_t)}, \tag{54}$$

and the last term can be bounded by using the Cauchy–Schwartz inequality as follows:

$$\left\langle \nabla \log \frac{\mu_t}{\pi}, \left( \sum_{i=1}^{\infty} \lambda_i \phi_i(x) \phi_i(\cdot) - P_{\mu_t, k} \right) \nabla \log \frac{\mu_t}{\pi} \right\rangle_{L^2(\mu_t)}$$

$$\leq \left\| \sum_{i=1}^{\infty} \lambda_i \phi_i(x) \phi_i(\cdot) - P_{\mu_t, k} \right\|_{\text{op}} \left\| \nabla \log \frac{\mu_t}{\pi} \right\|_{L^2(\mu_t)}^2. \tag{55}$$

Eq. (55) can be arbitrarily small since $\left\| \nabla \log \frac{\mu_t}{\pi} \right\|_{L^2(\mu_t)}^2$ is bounded (Lemma 2), the operator norm is bounded by HS norm, and property of the convergence in $L^2(\mu)$ norm. By setting $\| \sum_{i=1}^{\infty} \lambda_i \phi_i(x) \phi_i(\cdot) - P_{\mu_t, k} \|_{\text{op}} = \epsilon_0$, we have

$$\eta_t \left\langle \nabla \log \frac{\mu_t}{\pi}, (\epsilon_t I - P_{\mu_t, k}) \nabla \log \frac{\mu_t}{\pi} \right\rangle_{L^2(\mu_t)}$$

$$\leq \eta_t \left\langle \nabla \log \frac{\mu_t}{\pi}, \left( \epsilon_t I - \sum_{i=1}^{\infty} \lambda_i \phi_i(x) \phi_i(\cdot) \right) \nabla \log \frac{\mu_t}{\pi} \right\rangle_{L^2(\mu_t)} + \epsilon_0 \eta_t \left\| \nabla \log \frac{\mu_t}{\pi} \right\|_{L^2(\mu_t)}^2, \tag{56}$$

where $\epsilon_0$ is arbitrarily small and negligible for the convergence. Thus, it is sufficient to focus on the convergence of the spectral decomposition of $P_{\mu, k}$ in $L^2(\mu)$ norm.

Defining $v_t := \nabla \log \frac{\mu_t}{\pi}$ for simplicity in notation, we can obtain $v_t = \sum_{i=1}^{\infty} \langle v_t, \phi_i \rangle_{L^2(\mu)} \phi_i$ and $P_{\mu, k} v_t = \sum_{i=1}^{\infty} \lambda_i \langle v_t, \phi_i \rangle_{L^2(\mu)} \phi_i$ because the kernel function is dense in $L^2(\mu)$ and thus its eigenvectors are complete. Using these, we obtain

$$\epsilon_t \eta_t \left\| \nabla \log \frac{\mu_t}{\pi} \right\|_{L^2(\mu_t)}^2 - \eta_t I_{\text{stein}}(\mu_t | \pi) = \eta_t \sum_{i=1}^{\infty} (\epsilon_t - \lambda_i) \langle v_t, \phi_i \rangle_{L^2(\mu_t)}^2. \tag{57}$$

By setting $\epsilon_t$ sufficiently small, there exists a index $1 < j$ such that $\lambda_j > \epsilon_t > \lambda_{j+1}$. Hence, by regularizing $\{ \langle v_t, \phi_i \rangle_{L^2(\mu_t)}^2 \}_{i=1}^{\infty}$, we can render the left-hand side of Eq. (57) negative. For that purpose, we focus on the RKHS associated with $k$ given as

$$\mathcal{H} = \left\{ f \in L^2(\mu) \mid f = \sum_{k=1}^{\infty} a_i \phi_i, \sum_{i=1}^{\infty} \frac{\|a_i\|^2}{\lambda_i}, a_i \in \mathbb{R} \right\}, \tag{58}$$

where $\mathcal{H}$ is dense in $L^2(\mu)$. Thanks to this property of $\mathcal{H}$, there exists a function $v^{(l)} \in \mathcal{H}$ such that the sequence of $v^{(l)} \to v$ as $l \to \infty$ in $L^2(\mu)$ norm. We express such $v^{(l)}$ as

$$v_t^{(l)} = \sum_{i=1}^{\infty} b_i^{(l)} \phi_i \in \mathcal{H} \tag{59}$$

Note that we have that $\sum_{i=1}^{\infty} \frac{\|b_i^{(l)}\|^2}{\lambda_i} < \infty$ since $v_t^{(l)} \in \mathcal{H}$.

We consider replacing $v_t$ by $v_t^{(l)}$ to control the coefficient of $\{\phi_n\}$. By definition, for any $\epsilon > 0$, we can obtain $\|v_t - v_t^{(l)}\|_{L^2(\mu_t)} < \epsilon$ by choosing sufficiently large $l$. According to this fact, we obtain

$$\begin{aligned}
&\left\langle \nabla \log \frac{\mu_t}{\pi}, (\epsilon_t I - P_{\mu_t,k}) \nabla \log \frac{\mu_t}{\pi} \right\rangle_{L^2(\mu_t)} \\
&= \left\langle v_t - v_t^{(l)} + v_t^{(l)}, (\epsilon_t I - P_{\mu_t,k}) v_t - v_t^{(l)} + v_t^{(l)} \right\rangle_{L^2(\mu_t)} \\
&= \left\langle v_t^{(l)}, (\epsilon_t I - P_{\mu_t,k}) v_t^{(l)} \right\rangle_{L^2(\mu_t)} + 2 \left\langle v_t^{(l)}, (\epsilon_t I - P_{\mu_t,k}) v_t - v_t^{(l)} \right\rangle_{L^2(\mu_t)} \\
&\qquad\qquad + \left\langle v_t - v_t^{(l)}, (\epsilon_t I - P_{\mu_t,k}) v_t - v_t^{(l)} \right\rangle_{L^2(\mu_t)} \\
&= \left\langle v_t^{(l)}, (\epsilon_t I - P_{\mu_t,k}) v_t^{(l)} \right\rangle_{L^2(\mu_t)} + 2\epsilon_t \epsilon \|v_t\|_{L^2(\mu_t)}^2 + \epsilon_t \epsilon^2
\end{aligned} \tag{60}$$

According to Lemma 2, $\|v_t\|_{L^2(\mu_t)}^2$ is bounded for $t < \infty$, and the second and the third term can be arbitrarily small by $\epsilon$. Thus, we now focus on the term $\left\langle v_t^{(l)}, (\epsilon_t I - P_{\mu_t,k}) v_t^{(l)} \right\rangle_{L^2(\mu_t)}$ to analyze $\left\langle \nabla \log \frac{\mu_t}{\pi}, (\epsilon_t I - P_{\mu_t,k}) \nabla \log \frac{\mu_t}{\pi} \right\rangle_{L^2(\mu_t)}$.

From the above augments and the fact in Eq. (59), we can be rewritten Eq. (57) as

$$\epsilon_t \eta_t \left\| \nabla \log \frac{\mu_t}{\pi} \right\|_{L^2(\mu_t)}^2 - \eta_t I_{\text{stein}}(\mu_t|\pi) = \eta_t \sum_{i=1}^{\infty} (\epsilon_t - \lambda_i)(b_i^{(l)})^2 + o(\epsilon), \tag{61}$$

where the residual term $o(\epsilon)$, which comes from Eq. (60), can be arbitrarily small and thus it is trivial in our discussion.

Next, we consider to choose appropriate $\epsilon_t$. Recall that, by setting $\epsilon_t$ sufficiently small, there exists a index $1 < j$ such that $\lambda_j > \epsilon_t > \lambda_{j+1}$. Then, we can expand the right-hand side term of Eq. (61) as

$$\sum_{i=1}^{\infty} (\epsilon_t - \lambda_i)(b_i^{(l)})^2 = -\underbrace{\sum_{i=1}^{j} (\lambda_i - \epsilon_t)(b_i^{(l)})^2}_{=:A} + \underbrace{\sum_{i=j+1}^{\infty} (\epsilon_t - \lambda_i)(b_i^{(l)})^2}_{=:B}, \tag{62}$$

where $A, B > 0$ by definition. We now show that there exists $\epsilon_t > 0$ such that

$$\sum_{i=1}^{\infty} (\epsilon_t - \lambda_i)(b_i^{(l)})^2 = -A + B < 0. \tag{63}$$

This can be easily confirmed by the definition of $b_i^{(l)}$. Since $v_t^{(l)} \in \mathcal{H}$, we have $\sum_{i=1}^{\infty} \frac{\|b_i^{(l)}\|^2}{\lambda_i} < \infty$. This implies that $\sup_{i \le m}(b_m^{(l)})^2$ goes to 0 at least as the same order as $\lambda_i$. By setting $\epsilon_t$ sufficiently small, the corresponding index $j$ becomes large and then $\{(b_i^{(l)})^2\}_{i \ge j+1}$ becomes small. With this procedure, we obtain

$$\epsilon_t \eta_t \left\| \nabla \log \frac{\mu_t}{\pi} \right\|_{L^2(\mu_t)}^2 - \eta_t I_{\text{stein}}(\mu_t|\pi) < 0. \tag{64}$$

This implies that $\delta_t = 0$ in our AGF.

Finally, we show that we can choose $\epsilon_t$ so as to be independent of $t$. We would like to find the smallest index $m \in \mathbb{N}$ such that

$$\sum_{i=1}^{m} (b_i^{(l)})^2 > \sum_{i=m+1}^{\infty} (b_i^{(l)})^2, \tag{65}$$

holds. Thus, by definition

$$\sum_{i=1}^{m-1} (b_i^{(l)})^2 \leq \sum_{i=m}^{\infty} (b_i^{(l)})^2, \tag{66}$$

holds. Actually, we can find such index if we pick up sufficiently large $m$ because $\sup_{i \leq m}(b_m^{(l)})^2$ goes to 0 at least as the same order as $\lambda_i$ from $\sum_{i=1}^{\infty} \frac{\|b_i^{(l)}\|^2}{\lambda_i} < \infty$. Under such $m$, we obtain

$$\begin{aligned}
\sum_{i=1}^{\infty} (\epsilon_t - \lambda_i)(b_i^{(l)})^2 &= -\sum_{i=1}^{m} (\lambda_i - \epsilon_t)(b_i^{(l)})^2 + \sum_{i=m+1}^{\infty} (\epsilon_t - \lambda_i)(b_i^{(l)})^2 \\
&\leq -\sum_{i=1}^{m} \lambda_i (b_i^{(l)})^2 + 2\epsilon_t \sum_{i=1}^{m} (b_i^{(l)})^2 \\
&\leq (-\lambda_m + 2\epsilon_t) \sum_{i=1}^{m} (b_i^{(l)})^2.
\end{aligned} \tag{67}$$

By setting $\epsilon_t \leq \lambda_m/2$ and taking $o(\epsilon)(> 0)$ into account, we have the relationship as in Eq. (65).

If $m$ monotonically increases with respect to some sequence of $t$, the $m$-th eigenvalue $\lambda_m$ would become smaller as $t$ increases, leading to a decreasing upper bound of $\epsilon_t \leq \lambda_m/2$ as $t$ increases. To ensure convergence with the LSI, we must demonstrate that $m$ satisfying Eq. (65) and (66) does not monotonically increase with respect to the iteration $t$. This can be shown through a proof by contradiction as follows.

We assume that there exists a subsequence of $t$, $\{t_k\}_{k=1}^{\infty}$, such that $m$ satisfying Eq. (65) and (66) monotonically increases. Also, for any $k$, let $m$ corresponding to $t_k$ be denoted as $m_{t_k}$. Furthermore, we define the right-hand side of Eq. (66) as $S_m$, where $S_m := \sum_{i=m}^{\infty} (b_i^{(l)})^2$. From the definition of the index $m$, we have $m_{t_1} < m_{t_2} < \cdots < m_{t_k} < \ldots$, for all $k$. In this case, there exists a sequence in $S_{m_{t_1}}, \ldots, S_{m_{t_k}}, \ldots$ that goes to 0 because $\sup_{i \leq m}(b_m^{(l)})^2$ goes to 0 at least as the same order as $\lambda_i$ from $\sum_{i=1}^{\infty} \frac{\|b_i^{(l)}\|^2}{\lambda_i} < \infty$. However, a contradiction arises since $S_{m_{t_k}}$ approaching 0 does not satisfy Eq. (66).

This contradiction suggests that $m$ might increase for some range of $t$, but it is upper bounded with respect to $t$. This allows us to identify a largest $m$ as $m'$ that does not depend on $t$. Recalling Assumption 6, which provides a strictly positive lower bound for $\lambda_i$ denoted as $\hat{\lambda}_{m'}$, we see that this lower bound is also independent of $t$. Combining the above discussions, we can establish that $\epsilon_t$ can be upper-bounded by $\hat{\lambda}_{m'}$, meaning that $\epsilon_t \leq \hat{\lambda}_{m'}$. This implies that we can set a positive constant $c_0$ as an upper bound for $\epsilon_t$, and this constant is independent of $t$.

In conclusion, we have

$$\mathrm{KL}(\mu_{t+1}|\pi) \leq \mathrm{KL}(\mu_t|\pi) - c_0 \eta_t \left\| \nabla \log \frac{\mu_t}{\pi} \right\|_{L^2(\mu_t)}^2 \leq (1 - \tilde{c}_0 \eta_t) \mathrm{KL}(\mu_t|\pi), \tag{68}$$

where we used the LSI and summarized the LSI constants in $\tilde{c}_0$. By recursively applying the above inequality, we obtain

$$\mathrm{KL}(\mu_T|\pi) \leq \prod_{t=1}^{T-1} (1 - c_0 \eta_t) \mathrm{KL}(\mu_0|\pi). \tag{69}$$

We final note that if we have $\eta_t = c_1/T$,

$$\prod_{t=1}^{T}\left(1 - \frac{c_0 c_1}{t}\right) \leq \left(1 - \frac{1}{T}\sum_{t=1}^{T}\frac{c_0 c_1}{t}\right)^T \leq e^{-\sum_{t=1}^{T}\frac{c_0 c_1}{t}} \leq e^{-c_0 c_1 \log T} \leq \frac{1}{T^{c_0 c_1}}. \tag{70}$$

This concludes the proof.

### C.2 Proof of Lemma 2

For simplicity, we express $\eta_t$ as $\gamma$ is this proof. Let us define the mapping $\rho_s := \phi_s \# \mu_t$ for $s \in [0, \gamma]$ with $\phi_s := I - s v_t$ and $v_t = P_{\mu,k}\nabla \log \frac{\mu_t}{\pi}$. Then, by the change of variable formula, we have

$$\rho_s = |J\phi_s(\phi_s^{-1}(x))|^{-1}\mu_t(\phi_s^{-1}(x)), \tag{71}$$

where $\rho_\gamma = \mu_{t+1}$.

Our goal is to bound the following equality:

$$\left\|\nabla \log \frac{\rho_s}{\pi}\right\|_{L^2(\rho_s)}^2 = \int \left\langle \nabla \log \frac{\rho_s}{\pi}(x), \nabla \log \frac{\rho_s}{\pi}(x)\right\rangle \mathrm{d}\rho_s(x)$$

$$= \int \left\langle \nabla \log \frac{\mu_t(x)|J\phi_s(x)|^{-1}}{\pi(\phi_s(x))}, \nabla \log \frac{\mu_t(x)|J\phi_s(x)|^{-1}}{\pi(\phi_s(x))}\right\rangle \mathrm{d}\mu_t(x). \tag{72}$$

First, we apply the mean value theorem (as known as Taylor expansion of order 1) to $\psi(s) :=$ $\nabla \log \frac{\mu_t(x)|J\phi_s(x)|^{-1}}{\pi(\phi_s(x))}$ as a function of $s$. According to this theorem, there exists a constant $c \in [0, \gamma]$ such that

$$\psi(s) = \psi(0) + \gamma \frac{\mathrm{d}}{\mathrm{d}s}\psi(s)\Big|_{s=c}. \tag{73}$$

This implies

$$\left\|\nabla \log \frac{\mu_{t+1}}{\pi}\right\|_{L^2(\rho_{t+1})} = \left\|\nabla \log \frac{\rho_\gamma}{\pi}\right\|_{L^2(\rho_\gamma)} = \left\|\nabla \log \frac{\mu_t(x)|J\phi_s(x)|^{-1}}{\pi(\phi_s(x))}\right\|_{L^2(\mu_t)}$$

$$\leq \left\|\nabla \log \frac{\mu_t}{\pi}\right\|_{L^2(\mu_t)} + \gamma \left\|\frac{\mathrm{d}}{\mathrm{d}s}\psi(s)\Big|_{s=c}\right\|_{L^2(\mu_t)}, \tag{74}$$

where we used the triangle inequality. The second term in the right-hand side of Eq. (74) can be expressed as

$$\left\|\frac{\mathrm{d}}{\mathrm{d}s}\psi(s)\right\|_{L^2(\mu_t)}$$

$$= \left\|-\frac{\mathrm{d}}{\mathrm{d}s}\nabla \log |I - s J v_t| + \frac{\mathrm{d}}{\mathrm{d}s}\nabla V(\phi_s(x))\right\|_{L^2(\mu_t)}$$

$$= \left\|\nabla \mathrm{Tr}\left[(J\phi_s(x))^{-1}J v_t(x)\right] + \nabla \left\langle \nabla V(\phi_s(x)), v_t(x)\right\rangle\right\|_{L^2(\mu_t)}$$

$$= \left\|\nabla \sum_{ij}\left((J\phi_s(x))^{-1}\right)_{ij}\left(J v_t(x)\right)_{ji} + \nabla^2 V(\phi_s(x))v_t(x) + J v_t(x)\nabla V(\phi_s(x))\right\|_{L^2(\mu_t)}, \tag{75}$$

where $\nabla = (\partial_1, \ldots, \partial_d)^\top$ and $v_t = (v_1, \cdots, v_d)^\top$. For derivation, we first swap the time derivative and the gradient, and using the time derivative of $\Psi(s)$ that is shown in Appendix B of Salim et al. (2022) and Sun et al. (2023).

To bound Eq. (75), we use the following existing bounds. From Appendix B and Lemma C.1. in Salim et al. (2022), we have

$$\|v_t(x)\| \le B\|v_t\|_{\mathcal{H}} \le BC_1, \tag{76}$$
$$\|v_t\|_{\mathcal{H}} \le BC_1 \tag{77}$$

$$C_1 := \left(1 + 2L\sqrt{\frac{2\mathrm{KL}(\mu_0|\pi)}{C_{\mathrm{LS}}}} + LW_2(\mu_0, \delta_{x^*})\right) \tag{78}$$

where $x^* = \arg\min_{x \in \mathcal{X}} V(x)$. In Salim et al. (2022), they assumed that $T_1$ inequality holds for $\pi$, whereas we assumed that the LSI holds. Since $T_1$ inequality is satisfied if the LSI is available, their bound also holds in our setting. Also, from Appendix B in Salim et al. (2022), we have

$$\|Jv_t(x)\|_{\mathrm{HS}}^2 \le B^2\|v_t\|_{\mathcal{H}}^2 \le B^4C_1^2. \tag{79}$$

Moreover, from the proof of Lemma C.1. in Salim et al. (2022), for any $\mu \in \mathcal{P}_2(\mathcal{X})$, we have

$$\|\nabla V\|_{L^2(\mu)} \le 2L\sqrt{\frac{2\mathrm{KL}(\mu_0|\pi)}{C_{\mathrm{LS}}}} + LW_2(\mu_0, \delta_x^*) =: C_2. \tag{80}$$

From Appendix B in Salim et al. (2022), we have

$$\left\|(J\phi_s(x))^{-1}\right\|_{\mathrm{HS}} \le \alpha. \tag{81}$$

where this $\alpha$ is defined in the Assumption of Theorem 1. Using this result, the following relation holds

$$\left\|(J\phi_s(x))^{-2}\right\|_{\mathrm{HS}} \le \alpha^2. \tag{82}$$

In addition to these upper bounds (Eqs. (76)-(82)), we use the following fact:

$$\sum_{i,j,k=1}^d (\partial_i\partial_j v_k(x))^2 = \sum_{i,j,k=1}^d \langle\partial_i\partial_j k(x,\cdot), v_k\rangle_{\mathcal{H}_0}^2 = \sum_{i,j,k=1}^d \|\partial_i\partial_j k(x,\cdot)\|_{\mathcal{H}_0}^2\|v_k\|_{\mathcal{H}_0}^2$$
$$= \sum_{i,j=1}^d \|\partial_i\partial_j k(x,\cdot)\|_{\mathcal{H}_0}^2\|v_t\|_{\mathcal{H}}^2$$
$$\le B^2\|v_t\|_{\mathcal{H}}^2 \le B^4C_1^2, \tag{83}$$

where we used Assumption 3 in the last line.

To upper bound Eq. (75), we apply triangle inequality. Then we focus on the square of the first term in Eq. (75),

$$\sum_k (\partial_k(\sum_{ij} \left((J\phi_s(x))^{-1}\right)_{ij} (Jv_t(x))_{ji}))^2$$
$$= \sum_k ((\sum_{ij} \left((J\phi_s(x))^{-2}\right)_{ij} \partial_k(-tJv_t(x))_{ij} (Jv_t(x))_{ji}) + (\sum_{ij} \left((J\phi_s(x))^{-1}\right)_{ij} \partial_k (Jv_t(x))_{ji}))^2$$
$$\le \sum_k ((\sum_{ij} \alpha^2|\partial_k(-tJv_t(x))_{ij}|B^2C_1 + (\alpha|\partial_k (Jv_t(x))_{ji}|))^2$$
$$\le \sum_k ((\sum_{ij} (\alpha + t\alpha^2B^2C_1)|\partial_k(Jv_t(x))_{ij}|)^2$$
$$\le (\alpha + t\alpha^2B^2C_1)^2 d^2 \sum_{ijk} (\partial_k(Jv_t(x))_{ij})^2$$
$$\le (\alpha + t\alpha^2B^2C_1)^2 d^2 B^2\|v_t\|_{\mathcal{H}}^2$$
$$\le (\alpha + \gamma\alpha^2B^2C_1)^2 d^2 B^4C_1^2 =: D_1^2 \tag{84}$$

where $D_1 > 0$ and it is not depend on $t$.

Now we are ready for bounding Eq. (75). First, the second and third term in Eq. (75) can be upper bounded by

$$
\begin{aligned}
&\left\|\nabla^2 V(\phi_s(x))v_t(x) + Jv_t(x)\nabla V(\phi_s(x))\right\|_{L^2(\mu_t)} \\
&\leq \left\|\nabla^2 V(\phi_s(x))v_t(x)\right\|_{L^2(\mu_t)} + \|Jv_t(x)\nabla V(\phi_s(x))\|_{L^2(\mu_t)} \\
&\leq \left\|\nabla^2 V(\phi_s(x))\right\|_{\mathrm{op}}\|v_t(x)\|_{L^2(\mu_t)} + \|Jv_t(x)\|_{\mathrm{op}}\|\nabla V(\phi_s(x))\|_{L^2(\mu_t)} \\
&\leq LB\|v_t\|_{\mathcal{H}} + C_2\|v_t\|_{\mathcal{H}} \\
&\leq (LB^2 + C_2 B)C_1.
\end{aligned}
\tag{85}
$$

Substituting the upper bounds in Eqs. (84) and (85), we have

$$
\begin{aligned}
\left\|\frac{\mathrm{d}}{\mathrm{d}s}\psi(s)\right\|_{L^2(\mu_t)} &= \left\|-\frac{\mathrm{d}}{\mathrm{d}s}\nabla\log|I - sJv_t| + \frac{\mathrm{d}}{\mathrm{d}s}\nabla V(\phi_s(x))\right\|_{L^2(\mu_t)} \\
&\leq ((\alpha + \gamma\alpha^2 B^2 C_1)dB + LB + C_2)\|v_t\|_{\mathcal{H}} \\
&\leq ((\alpha + \gamma\alpha^2 B^2 C_1)dB + LB + C_2)BC_1 =: C_3.
\end{aligned}
\tag{86}
$$

Thus,

$$
\begin{aligned}
\left\|\nabla\log\frac{\mu_{t+1}}{\pi}\right\|_{L^2(\mu_{t+1})} &= \left\|\nabla\log\frac{\rho_\gamma}{\pi}\right\|_{L^2(\rho_\gamma)} \\
&\leq \left\|\nabla\log\frac{\mu_t}{\pi}\right\|_{L^2(\mu_t)} + \gamma((\alpha + \gamma\alpha^2 B^2 C_1)dB + LB + C_2)\|v_t\|_{\mathcal{H}} \\
&\leq \left\|\nabla\log\frac{\mu_t}{\pi}\right\|_{L^2(\mu_t)} + \gamma C_3
\end{aligned}
\tag{87}
$$

By induction, we have the following results:

$$
\left\|\nabla\log\frac{\mu_T}{\pi}\right\|_{L^2(\mu_T)} \leq \left\|\nabla\log\frac{\mu_0}{\pi}\right\|_{L^2(\mu_0)} + \sum_{t=0}^{T-1}\gamma_t C_3.
\tag{88}
$$

This completes the proof.

## D  Details of experimental settings

We set the target distribution as $p(x) = \mathcal{N}(x|\mu^*, \Sigma^*)$ with $\mu^* = [1, 1]$ and $\Sigma^* = \mathrm{diag}(1, 1)$, where diag is a diagonal matrix. For the Gaussian mixture distribution, we set $p(x) = \frac{2}{3}\mathcal{N}(x|\mu_1^*, \Sigma_1^*) + \frac{1}{3}\mathcal{N}(x|\mu_2^*, \Sigma_2^*)$ with $\mu_1^* = [2, -2]$, $\mu_2^* = [-2, 2]$, $\Sigma_1^* = \Sigma_2^* = \mathrm{diag}(1, 1)$, which is the extension of the experimental settings in Liu & Wang (2016) for the two-dimensional setting. We generated the initial particles from $\mathcal{N}(x|\mu_0, \Sigma_0)$ with $\mu_0 = [0, 0]$ and $\Sigma_0 = \mathrm{diag}(1, 1)$ or $\mu_0 = [-5, -5]$ and $\Sigma_0 = \mathrm{diag}(1, 1)$ for the Gaussian and the Gaussian mixture experiments, respectively.

We adopted the RBF kernel $k(x, y) = \exp(\frac{1}{h}\|x - x'\|_2^2)$, which is commonly used in practice and satisfies the assumptions in Section 4. The bandwidth $h$ was selected by the median trick, i.e., $\mathrm{med}^2/\log n$ as in Liu & Wang (2016), where med is the median of the pairwise distance between the current particles.

As we stated in Section 5, we simply set the decaying step size $\gamma_t = 1/(1 + t^\beta)(= \mathcal{O}(1/t^\beta))$ suggested by Theorem 1 and did not use the Adagrad-based stepsize, which is adopted in related studies such as Korba et al. (2021) and others. We set the initial stepsize as $\gamma_0 = 0.01$ for all experiments. We evaluated the KL divergence: $\mathrm{KL}(\mu_T|\pi)$ and the cumulative mean of KSD: $\frac{1}{T}\sum_{t=1}^{T} I_{\mathrm{stein}}(\mu_t|\pi)$, which are theoretically guaranteed sub-linear convergence.

We conducted our experiments based on the above settings using $\{5, 10, 100, 1000\}$ particles.

# E    Additional experimental results

In this section, we provide the additional experimental results.

We confirmed in Section 5 that SVGD with the RBF kernel tends to achieve sub-linear convergence both in $\mathrm{KL}(\mu_T|\pi)$ and in $\frac{1}{T}\sum_{t=1}^{T}I_{\mathrm{stein}}(\mu_t|\pi)$ (see Figures 3 and 4). As for the Gaussian mixture settings, we can observe the same behavior (see Figures 5 and 6), which also supports Theorem 1.

As discussed in Appendix A, the bias in the KL divergence persists as we increase the value of $T$ because we utilized a finite number of particles, leading to $\delta_t \neq 0$ in AGF. Such a bias can be reduced by increasing the number of particles increases (see Figures 3-6, 7, and 8). On the flip side, even in the Gaussian mixture experiments, using a large number of particles results in slower convergence for both the KSD and KL divergence.

This phenomenon can be attributed to the existence of extremely small eigenvalues of $P_{\mu,k}$ when a larger number of particles is used, as the eigenvalues of the RBF kernel decay exponentially fast (Wainwright, 2019). To confirm this fact, we measured the eigenvalues of the Gram matrix obtained from the RBF kernel function at three points: the initial stage of learning ($t = 1$), the midpoint ($t = 5 * 10^4$), and the final stage ($t = 10^5$). We summarize these results in Figures 9 and 10. We can see that the exponential decay of the eigenvalues tends to be occurring as the number of particles increases.

Assumption 6, which states that the eigenvalues have a strictly positive lower bound and upper bound that are independent of $t$, is crucial for showing the sub-linear convergence of SVGD in KL divergence under the setting of an infinite number of particles. Since it is difficult to theoretically show this fact, we instead conducted numerical experiments to confirm that the dependence of the upper bound (maximum value) and lower bound (minimum value) of the eigenvalues on the variable $t$ diminishes as the number of particles increases. As a metric for measuring time-dependence, we employed the following growth rate for the time interval $[t_1, t_2](t_2 > t_1)$:

$$\frac{|\tilde{\lambda}_{t_2} - \tilde{\lambda}_{t_1}|}{t_2 - t_1}, \quad \frac{|\bar{\lambda}_{t_2} - \bar{\lambda}_{t_1}|}{t_2 - t_1},$$

where $\{\tilde{\lambda}_{t_j}, \bar{\lambda}_{t_j}\}$ ($j = 1, 2$) is the maximum and minimum value of the eigenvalues at iteration $t_j$. In the above metric, if the dependence of the eigenvalues for $t$ is small, meaning that the changes in $\{\tilde{\lambda}_{t_j}, \bar{\lambda}_{t_j}\}$ with the progression of $t$ are small, then the value will be small. Furthermore, when the overall behavior indicates minimal fluctuation in this value throughout the training process, it signifies that the changes in the eigenvalues with the progression of $t$ are small. This, in turn, reflects the small dependence of $\{\tilde{\lambda}_{t_j}, \bar{\lambda}_{t_j}\}$ with respect to $t$.

We summarized the experimental results under the following three case: $(t_1, t_2) = (0, 5 * 10^4)$ (refer to case (i)), $(5 * 10^4, 10^5)$ (refer to case (ii)), and $(0, 10^5)$ (refer to case (iii)) in Figures 11 and 12. To begin with, we can see that the change between the midpoint and endpoint of $t$, i.e., the case (ii), yields the smallest value for all particle settings. This suggests that during this period, SVGD is gradually approaching convergence, which seems to align with the results of $\frac{1}{T}\sum_{t=1}^{T}I_{\mathrm{stein}}(\mu_t|\pi)$ in Figures 3-6. On the other hand, in the low number of particles setting, there is a significant difference in the amount of change for each configuration, whereas when a large number of particles are used, this difference becomes small. This key finding—that the fluctuations of both maximum and minimum eigenvalues with respect to $t$ decrease as the number of particles increases—provides empirical backing for Assumption 6. The trend strongly suggests that in the infinite-particle limit ($m \to \infty$), the particle distribution $\mu_t$ stabilizes as it converges, and consequently, the eigenvalue spectrum of the operator $P_{\mu,k}$ becomes independent of time $t$. This supports that there is a case when our use of Assumption 6 is valid.

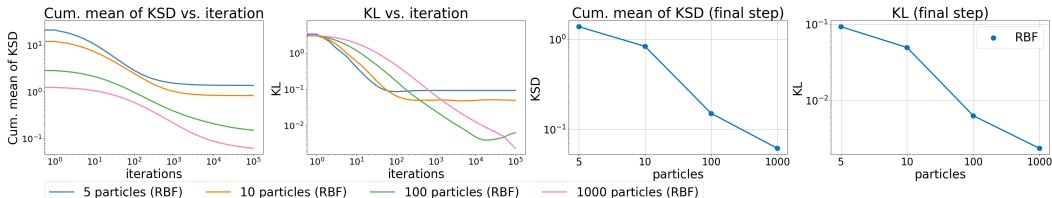

Figure 5: The convergence behavior in terms of $\mathrm{KL}(\mu_T|\pi)$ and $\frac{1}{T}\sum_{t=1}^{T} I_{\mathrm{stein}}(\mu_t|\pi)$ for all $T$ under two-dimensional Gaussian mixture experiments ($\beta = 0.67 \approx 2/3$).

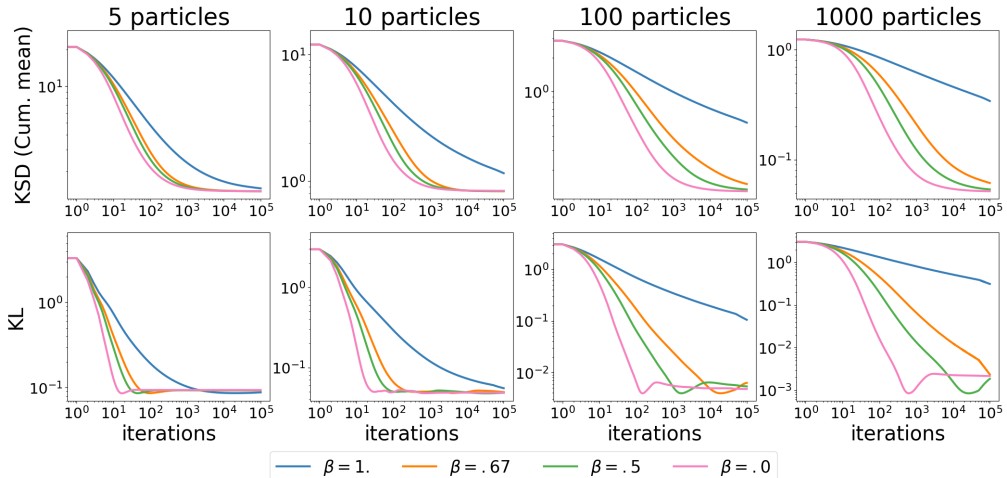

Figure 6: Convergence in $\mathrm{KL}(\mu_T|\pi)$ and $\frac{1}{T}\sum_{t=1}^{T} I_{\mathrm{stein}}(\mu_t|\pi)$ for all $T$ under different particles and stepsize settings ($\beta = \{0., 0.5, 0.67, 1.\}$).

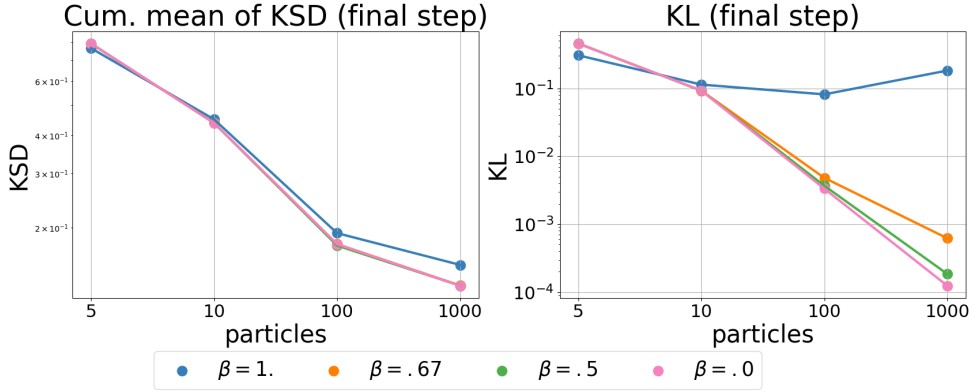

Figure 7: Change in convergence for variations in the order of stepsize with $\beta = \{0., 0.5, 0.67, 1.\}$ under Gaussian distribution estimation experiments.

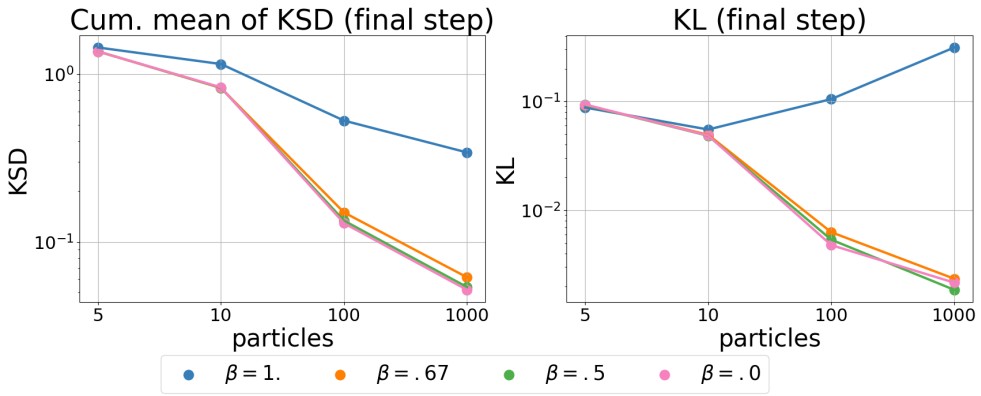

Figure 8: Change in convergence for variations in the order of stepsize with $\beta = \{0., 0.5, 0.67, 1.\}$ under Gaussian distribution estimation experiments.

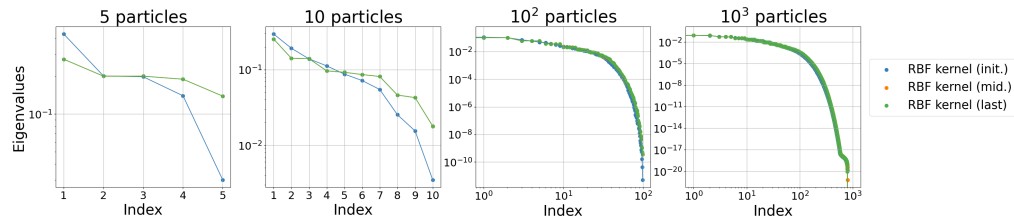

Figure 9: Eigenvalues of the Gram matrix in the two-dimensional Gaussian distribution experiments ($\beta = 0.67 \approx 2/3$).

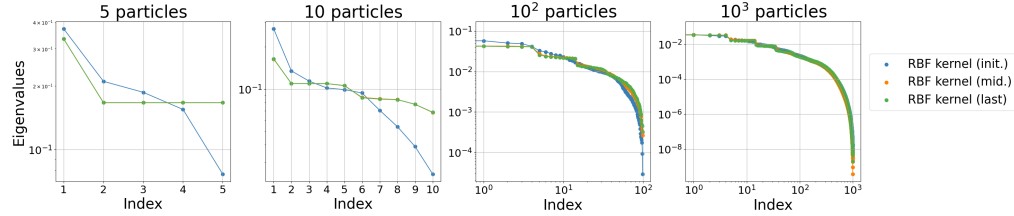

Figure 10: Eigenvalues of the Gram matrix in the two-dimensional Gaussian mixture experiments ($\beta = 0.67 \approx 2/3$).

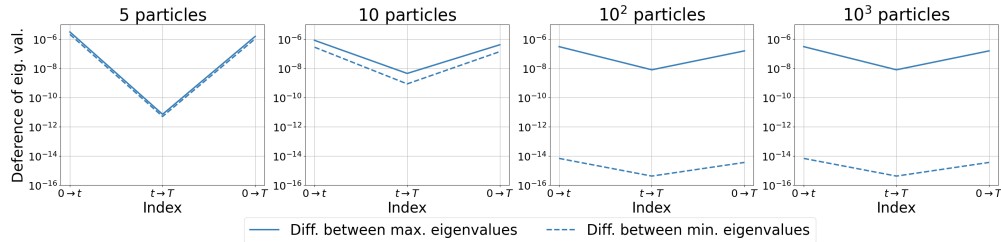

Figure 11: Difference between {maximum, minimum} eigenvalues of the Gram matrix in the two-dimensional Gaussian distribution experiments ($\beta = 0.67 \approx 2/3$). In this figure, $(t, T)$ represents $(5 * 10^4, 10^5)$.

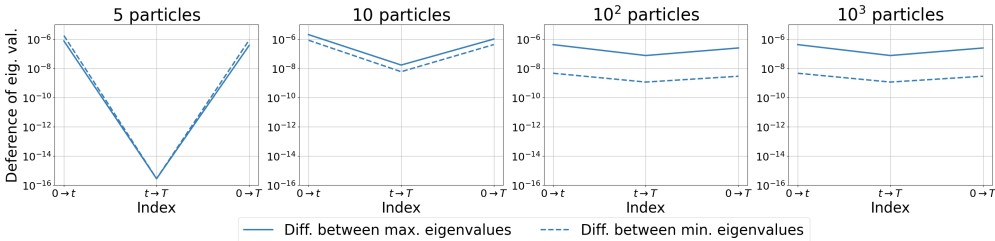

Figure 12: Difference between {maximum, minimum} eigenvalues of the Gram matrix in the two-dimensional Gaussian mixture experiments ($\beta = 0.67 \approx 2/3$). In this figure, $(t, T)$ represents $(5 * 10^4, 10^5)$.

# F Additional Experiments on a Real-World Dataset

This section details the experiments conducted to evaluate the performance of SVGD with a finite number of particles on a real-world dataset. We describe the model specification, the procedure for generating reference posterior samples via MCMC for evaluation, and the specific configurations of the SVGD experiments.

## F.1 Model: Hierarchical Bayesian Logistic Regression

The model used is a Hierarchical Bayesian Logistic Regression for binary classification. The goal is to infer the posterior distribution of the regression coefficients $\boldsymbol{\beta} \in \mathbb{R}^D$ and a global precision parameter $\tau > 0$.

**Parameterization**  To ensure the positivity constraint on the precision parameter $\tau$ and to improve the stability of the inference process, we reparameterize it by using its logarithm. The inference is performed on the parameter vector $\boldsymbol{\theta} = [\boldsymbol{\beta}^T, \phi]^T \in \mathbb{R}^{D+1}$, where $\phi = \log \tau$.

**Model Hierarchy**  The model is defined by the following hierarchy:

- **Likelihood**: The binary labels $y_n \in \{0, 1\}$ for each observation $\boldsymbol{x}_n$ are modeled by a Bernoulli distribution with a logistic link function (sigmoid function $\sigma(\cdot)$).

$$y_n | \boldsymbol{x}_n, \boldsymbol{\beta} \sim \text{Bernoulli}(\sigma(\boldsymbol{x}_n^T \boldsymbol{\beta}))$$

- **Priors**: A hierarchical prior structure is employed.

  - The regression coefficients $\boldsymbol{\beta}$ are drawn from a zero-mean Normal distribution, with precision $\tau = e^\phi$:

  $$\boldsymbol{\beta} | \phi \sim \mathcal{N}(\mathbf{0}, e^{-\phi/2} \mathbf{I}_D)$$

  - The precision parameter $\tau$ is drawn from a Gamma distribution:

  $$\tau \sim \text{Gamma}(\alpha_0, \beta_0)$$

  For the SVGD inference operating on $\phi = \log \tau$, the prior for $\phi$ is derived using the change of variables formula, $p(\phi) = p(\tau = e^\phi) \cdot |\mathrm{d}\tau/\mathrm{d}\phi| = p(e^\phi) \cdot e^\phi$.

## F.2 Reference Posterior Samples via MCMC

To evaluate the SVGD particles, we first generate a set of reference samples from the posterior distribution $p(\boldsymbol{\beta}, \tau | \mathbf{X}, \mathbf{y})$ using MCMC. These samples serve as an empirical representation of the true posterior. We use the No-U-Turn Sampler (NUTS), as implemented in `CmdStanPy`.

The sampling process involves running multiple independent chains in parallel. After discarding an initial set of warmup samples from each chain, the remaining post-warmup samples are combined. To ensure consistency with the SVGD parameterization, the samples of the precision parameter, $\tau^{\text{MCMC}}$, are log-transformed to $\phi^{\text{MCMC}} = \log \tau^{\text{MCMC}}$. The final collection of reference samples, denoted $\{\boldsymbol{\theta}_j^{\text{MCMC}}\}_{j=1}^{N_{\text{MCMC}}} = \{[\boldsymbol{\beta}_j^{\text{MCMC}}, \phi_j^{\text{MCMC}}]\}_{j=1}^{N_{\text{MCMC}}}$, is saved and used to compute various divergence measures, such as KSD and KL divergence estimated via KDE.

To validate that the MCMC samples form a reasonable approximation of the true posterior, we evaluated their predictive performance. The average log-likelihood over the posterior samples was $-0.5186$ per data point, while the mean prediction accuracy reached $75.57\%$ on the training data and $75.52\%$ on the test data. These performance metrics are consistent with results from related SVGD studies, such as Liu & Wang (2016) and Wang et al. (2019). This confirms that our MCMC procedure converged to a sensible posterior distribution, justifying the use of its samples as a reliable benchmark for our SVGD experiments.

Table 1: Experimental Setup

| Component | Parameter | Value |
|---|---|---|
| **Dataset** | Name | Covertype (UCI Repository) |
| | Preprocessing | Binary classification, pre-scaled features |
| | Data size for MCMC | 10,000 samples |
| | Splitting | 80% training, 20% testing |
| **Model** | Type | Hierarchical Bayesian Logistic Regression |
| | Parameterization | $\boldsymbol{\theta} = [\boldsymbol{\beta}^T, \log\tau]^T$ |
| | Feature dimensions ($D$) | 54 |
| | Total parameters ($D+1$) | 55 |
| **MCMC** | Sampler | NUTS (via `CmdStanPy`) |
| | Chains | 4 |
| | Samples per chain | 2,000 (post-warmup) |
| | Warmup per chain | 1,000 |
| | Prior Hyperparameters ($\alpha_0, \beta_0$) | (1.0, **0.1**) |
| **SVGD** | Iterations | 10,000 |
| | Number of particles ($N$) | Varied across $\{5, 10, 20, 50\}$ |
| | Optimizer | Gradient Ascent |
| | Base step size ($\epsilon_0$) | $1 \times 10^{-2}$ |
| | Decay factor ($d$) | 1.0 |
| | Decay exponent ($\beta$) | Varied across $\{0.0, 0.5, 0.67, 1.0\}$ |
| | Kernel | RBF with median heuristic |
| | Particle Initialization | $\boldsymbol{\beta} \sim \mathcal{N}(\mathbf{0}, 0.1\mathbf{I})$, $\phi = \log\tau \sim \mathcal{N}(\log(0.1), 0.1^2)$ |
| | Prior Hyperparameters ($\alpha_0, \beta_0$) | (1.0, **0.01**) |

### F.3   Inference via SVGD

We use SVGD to approximate the target posterior distribution. SVGD iteratively transports a set of particles $\{\boldsymbol{\theta}_i\}_{i=1}^N$ in the reparameterized space $\mathbb{R}^{D+1}$ to match the target distribution. The update for each particle $\boldsymbol{\theta}_i$ is a gradient ascent step:

$$\boldsymbol{\theta}_i \leftarrow \boldsymbol{\theta}_i + \epsilon_t \boldsymbol{\phi}(\boldsymbol{\theta}_i)$$

where $\epsilon_t$ is the learning rate at iteration $t$, and $\boldsymbol{\phi}(\cdot)$ is the velocity field.

We adopt the same kernel function and optimization algorithm in our experiments in Section 5 as follows. We use the RBF kernel $k(\boldsymbol{x}, \boldsymbol{y}) = \exp\left(-\frac{\|\boldsymbol{x}-\boldsymbol{y}\|^2}{2h^2}\right)$. The bandwidth $h$ is set at each iteration using the median heuristic. The particles are updated using standard gradient ascent. The learning rate $\epsilon_t$ is not fixed but follows a decay schedule, which is detailed in the experimental setup.

### F.4   Experimental Setup

The detailed experimental settings for the dataset, MCMC, and SVGD are summarized in Table 1. The learning rate at each iteration $t$ is determined by a decay schedule described in Appendix D.

### F.5   Results

In this section, we discuss the experimental results for the BLR model, which represents a more complex and practical scenario.

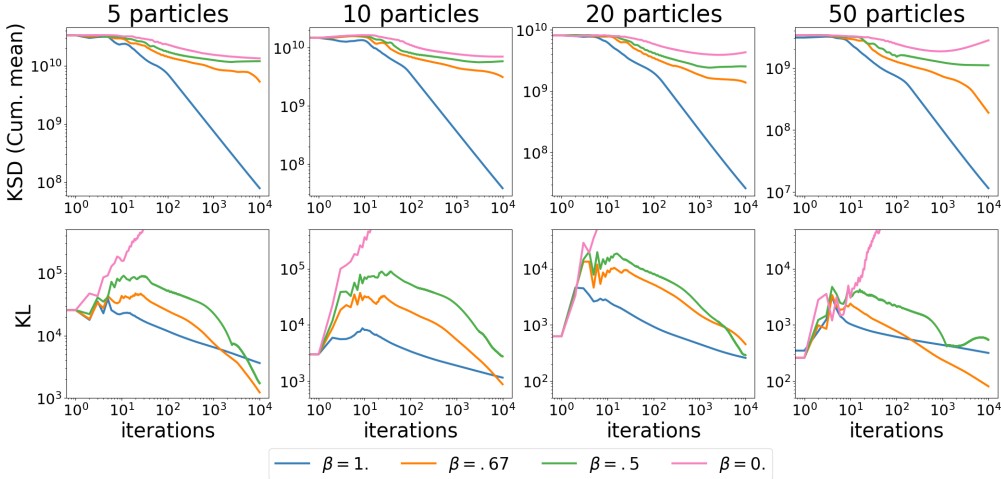

Figure 13: Convergence in $\mathrm{KL}(\mu_T|\pi)$ and $\frac{1}{T}\sum_{t=1}^{T} I_{\text{stein}}(\mu_t|\pi)$ for all $T$ under different particles and stepsize settings ($\beta = \{0., 0.5, 0.67, 1.\}$).

As shown in Figures 13 and 14, the BLR experiment also demonstrates that SVGD with an RBF kernel achieves sub-linear convergence for both the time-averaged KSD and the KL divergence. This aligns with the theoretical insights of Theorem 1 and confirms that this behavior is not limited to simpler target distributions.

Consistent with our previous findings, the bias in the KL divergence, stemming from the use of a finite number of particles, is evident. Figure 14 shows that this bias tends to diminish as the number of particles increases from 5 to 50 at least under the well-controlled stepsize. We also observe the same trade-off: while a larger number of particles leads to a better final approximation of the target distribution, it can result in slower initial convergence for both KSD and KL divergence, as seen in Figure 13.

This phenomenon is again attributable to the spectral properties of the RBF kernel's Gram matrix. We investigated the eigenvalues at the initial ($t = 1$), middle, and final stages of the learning process, with the results summarized in Figure 15. Just as in the simpler MVN and GM experiments, the eigenvalues exhibit exponential decay, and this decay becomes more pronounced as the number of particles increases. This leads to extremely small eigenvalues that can slow down the convergence of the particle system.

To provide further empirical support for Assumption 6—the time-independence of the eigenvalue bounds in the infinite-particle limit—we analyzed the growth rate of the maximum and minimum eigenvalues over time. The results for the BLR experiment are presented in Figure 16. The observations are threefold and mirror the previous experiments: (1) The rate of change of the eigenvalues is smallest in the latter half of the training (case (ii)), corresponding to the period where the algorithm approaches convergence; (2) The overall magnitude of the eigenvalue fluctuations tend to be smaller for a larger number of particles; (3) The difference in the rate of change between the early and late stages of training becomes less pronounced as the number of particles increases.

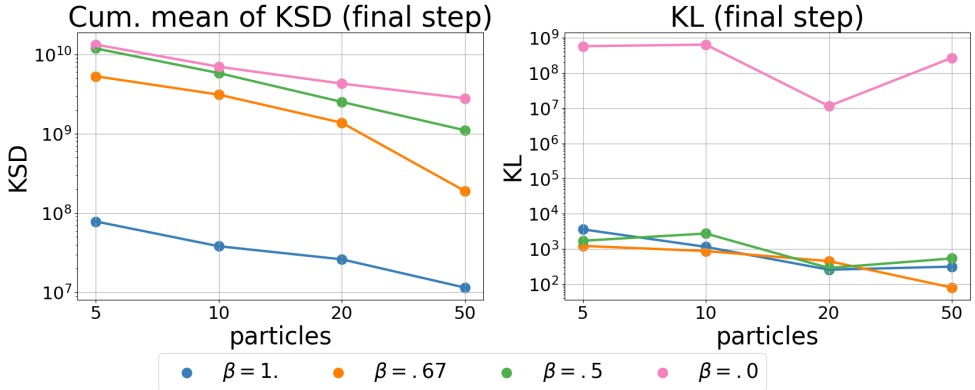

Figure 14: Change in convergence for variations in the order of stepsize with $\beta = \{0., 0.5, 0.67, 1.\}$ under Gaussian distribution estimation experiments.

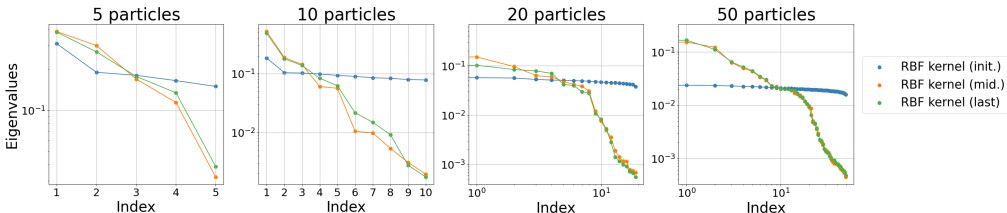

Figure 15: Eigenvalues of the Gram matrix in the Bayesian logistic regression experiments ($\beta = 0.67 \approx 2/3$).

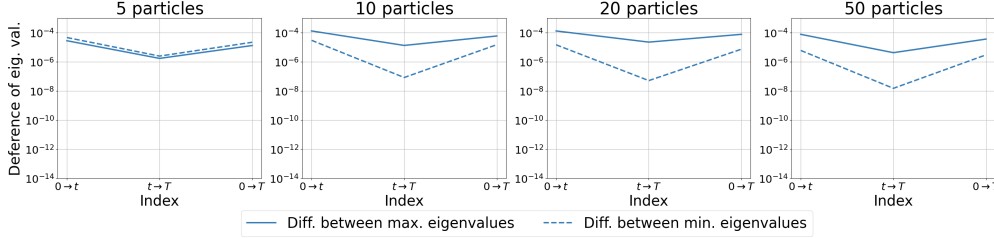

Figure 16: Difference between {maximum, minimum} eigenvalues of the Gram matrix in the Bayesian logistic regression experiments ($\beta = 0.67 \approx 2/3$). In this figure, $(t, T)$ represents $(5 * 10^4, 10^5)$.

