# OpenReview forum: "On the Convergence of SVGD in KL divergence via Approximate gradient flow"
_TMLR — Accepted by TMLR_

### Review · Reviewer_6MbM · 2025-06-20

**Summary Of Contributions:**

This paper provides a new theoretical perspective and convergence analysis for Stein Variational Gradient Descent (SVGD). In particular, the submission introduces a novel framework and proves convergence results that were previously unknown for SVGD.

Key contributions include:

1. (ϵ,δ)-Approximate Gradient Flow (AGF): The authors propose an (ϵ,δ)-AGF framework, which allows viewing SVGD as an approximate gradient flow in KL divergence. This framework extends ideas from Wasserstein gradient flows to quantify how closely SVGD follows the ideal KL-steepest descent direction.

2. Descent Inequality (SVGD Update Accuracy): Using the AGF framework, the paper derives a descent inequality that characterizes the alignment between the SVGD update direction and the true KL gradient.

3. KL Divergence Convergence Guarantee: Under appropriate conditions, the authors prove for the first time a convergence guarantee for SVGD in terms of KL divergence.

4. Empirical Validation (KL & KSD Decay): The authors validate their theory with numerical experiments on synthetic target distributions (e.g. multivariate Gaussian and Gaussian mixture targets). The experiments demonstrate that both the KL divergence and the kernel Stein discrepancy (KSD) of the particle approximation decrease over iterations, roughly following the sub-linear decay trends predicted by theory.

**Audience:**

Yes

**Broader Impact Concerns:**

There are no significant ethical or societal concerns apparent in this work. The contributions are theoretical and methodological, aiming to understand and improve the convergence of a sampling algorithm. As such, the research does not involve sensitive data or application-specific risks. The primary impact is to advance theory in variational inference, which is a positive scientific contribution. We encourage the authors to be mindful that any downstream use of SVGD (e.g. in applications like AI for healthcare or fairness-critical domains) should still be vetted for ethical considerations, but this recommendation is general. In summary, this submission does not present any direct ethical issues or harmful consequences in itself.

**Claims And Evidence:**

Yes

**Requested Changes:**

The paper is well-written and technically strong. Below are a few suggested changes or additions that could further improve it, though they are not strictly required:

1. Discuss Finite-Particle Implications: Even though a rigorous finite-m convergence analysis is left for future work, the authors should expand the discussion on how the (ϵ,δ)-AGF insights might extend to finite particle settings. For instance, clarifying the challenges (such as particles’ dependency in RKHS and how ϵ_t,δ_t might scale with m) and any conjectures or preliminary thoughts would be valuable to readers who wonder about the practical case. A brief subsection or remark on this would strengthen the impact.

2. Contextualize (or Relax) Assumption 6: Given that Assumption 6 is flagged as rather strong by the authors themselves, it would help to either weaken this assumption or better explain its necessity. The authors could discuss how realistic this condition is (do common kernels and distributions satisfy it? can it be empirically checked?) and what the intuition is behind it. Ideally, if possible, providing a version of the convergence result under a milder assumption (even at the cost of a slower rate or extra conditions) would make the theory more robust. If relaxing it is too difficult, clearly stating its role and any evidence that it approximately holds in experiments would be useful.

3. Empirical Estimation of δ_t: To complement the theory, the authors could consider an empirical analysis of the (ϵ_t,δ_t ) parameters. For example, in a toy scenario where the true gradient flow is known, one might measure the inner product -〈log(μ_t/π),g_(μ_t ) 〉 and the norm ‖∇log(μ_t/π)‖^2to estimate the effective ϵ_t and δ_t over iterations. Plotting how δ_t behaves in practice (with finite m) would provide insight into the accuracy of SVGD’s gradient approximation beyond the infinite-particle ideal.

4. Broaden the Experiments: The current experiments focus on relatively low-dimensional synthetic distributions (Gaussian mixtures, etc.). It would strengthen the paper to explore higher-dimensional or more realistic targets. For instance, testing SVGD on a Bayesian posterior from a real dataset (e.g. a UCI dataset or a simple Bayesian neural network) could demonstrate the method’s convergence properties in a practical setting. Even if the theory’s assumptions (like log-concavity) don’t strictly hold in those cases, seeing whether KL divergence still decreases (perhaps sub-linearly) would highlight the robustness and relevance of the findings. Such experiments would address potential concerns about how the theory translates to real-world applications.

**Strengths And Weaknesses:**

Strengths:
---The paper tackles an important theoretical gap – convergence of SVGD in KL divergence – with a novel and well-founded approach. The introduction of the (ϵ,δ)-AGF framework is innovative, extending gradient-flow theory to accommodate approximation error. This allows the authors to derive rigorous results where none existed before. Notably, the analysis yields the first provable sub-linear KL-convergence rate for SVGD, a significant theoretical advance. The proofs are technically detailed and appear rigorous, relying on reasonable assumptions (e.g. smooth RBF kernels, log-concavity) that make the results credible in practical scenarios.

---Another strength is the alignment between theory and experiment: the numerical results (tracking KL and KSD over time) strongly support the theoretical predictions, adding confidence that the theory is capturing relevant aspects of SVGD’s behavior. Overall, the paper’s contributions are novel, its theoretical insights are deep, and the experiments are well-designed to corroborate the claims.

Weaknesses:
---A limitation of the work is that the convergence theory is developed only for the infinite-particle (population) limit of SVGD. The provided guarantees hold as the number of particles m approaches infinity, whereas practical SVGD runs with finite m. Extending these results to finite particles remains an open challenge (acknowledged by the authors).

---Another weakness is the reliance on a strong assumption (referred to as Assumption 6 in the paper) regarding the kernel’s eigenvalues or related spectral quantities. This assumption was needed to ensure a certain constant c_0 remains of order 1 over time, but it is quite restrictive and difficult to verify in practice. It would be more reassuring if the theory could avoid or relax this condition, as currently it may limit the generality of the results.

---Finally, while the AGF framework introduces parameters ϵ_t and δ_t to quantify the approximation error per iteration, the analysis does not quantify δ_t explicitly (beyond ensuring it goes to 0 as m approaches infinity). In other words, we lack a clear sense of how large the per-step error term might be for finite m, or how fast it decays — this gap leaves some practical uncertainty in applying the theory.

---

> ### Author Response · Authors · 2025-07-08
> **Author responses**
>
> We thank Reviewer 6MbM for their sharp and insightful questions regarding our key assumptions and the practical implications of our framework. These questions have prompted us to add more detailed justifications to the manuscript.
>
> **Q1. Regarding the concern that Assumption 6 is overly strong.**
>
> **A.** We agree that Assumption 6 is strong, and we have revised Appendices A and E to better explain both its technical necessity and its empirical justification.
>
> - **Technical Necessity:** Assumption 6 is a crucial technical condition required to prove Theorem 1. As outlined in Section 4.2 and detailed in the full proof in Appendix C, our analysis aims to show that SVGD is a $(c_0,0)$-AGF, where $c_0$​ is a constant independent of the iteration time $t$. This constant $c_0$​ is derived from the eigenvalues ${\lambda_i​}$ of the kernel operator. If the eigenvalues were allowed to decay to zero as $t\to \infty$, then the relative approximation quality $\epsilon_t$ (which we show is a constant $c_0$​) would also decay. This would prevent us from applying the Log-Sobolev Inequality (LSI) to obtain the final sub-linear convergence rate. Thus, the assumption of time-invariant positive bounds on the eigenvalues is what allows us to establish a time-independent $\epsilon_t=c_{0}​ > 0$.
> - **Empirical Justification:** While a formal proof of Assumption 6 is challenging, we argue that it is an empirical idealization for the infinite-particle limit. We have expanded Appendix E with a detailed quantitative analysis to support this.
> Figures 11 and 12 in the appendix now present a study of the change in the maximum and minimum eigenvalues over the course of training.
> The key finding, as stated in the appendix, is that "with an increase in the number of particles, the fluctuations of $\overline{\lambda_{t_j}}, \underline{\lambda_{t_j}}$ w.r.t. $t$ decrease." This trend suggests that in the infinite-particle limit ($m\to \infty$), the particle distribution $\mu_t$​ stabilizes as it converges, and consequently, the eigenvalue spectrum of the operator $P_{\mu_{t}​,k}$​ becomes independent of time $t$. This supports that there is a case when our use of Assumption 6 is valid.
>
> **Q2. On the suggestion to empirically estimate the bias term $\delta_{t}$.**
>
> **A.** We agree that characterizing the bias term $\delta_t$​ is the critical next step for a complete understanding of practical, finite-particle SVGD, and we view this as an important future research direction.
>
> Our framework provides the precise theoretical origin of this bias term. As explained in Appendix A, in the finite-particle setting, the eigenvectors of the empirical kernel operator do not form a CONS for the true underlying space $L_{2}(\mu)$. This incompleteness creates a projection error when approximating the true Wasserstein gradient. In our AGF framework, this error manifests as the absolute bias term $\delta_t$​, which is equal to the squared $L_2$ norm of the gradient component residing in the null space of the kernel operator, i.e., $\delta_t = \|\Psi \|_{L^{2}(\mu)}^{2}$.
>
> However, empirically estimating this term is highly non-trivial. The error component $\Psi$ depends on the entire (and unknown) particle distribution $μ_t$​ and the target $\pi$. More importantly, it is complicated by the correlations between particles that are inherent to the SVGD update rule. As we state in our conclusion, the theoretical properties of this gradient approximation error for correlated particles are difficult to analyze without strong assumptions like specification of the models. A naive empirical estimation would be difficult to perform and interpret without a better theoretical handle on these complex dependencies.
>
> Therefore, we fully agree with the reviewer's sentiment. Our paper provides the theoretical groundwork—the AGF framework—that enables the precise definition of this bias term $\delta_t$. Characterizing and estimating it for finite, correlated particles is the "main challenge" for extending our work, as stated in Section 6. We have added a sentence to our conclusion to explicitly acknowledge that the empirical estimation of $\delta_t$​ is a valuable, albeit challenging, component of this future work.
>
> **Q3. On broadening the experiments to more realistic, higher-dimensional settings.**
>
> **A.** We thank the reviewer for this excellent and constructive suggestion. We completely agree that demonstrating the applicability of our findings in a more realistic, higher-dimensional setting would significantly enhance the impact of our work.
> We have already begun work to implement this suggestion and are currently running experiments on Bayesian Logistic Regression with some UCI dataset. These results will be fully incorporated into the final paper, at the latest, by the camera-ready version.
> If it is possible, we would be happy to update the manuscript with these findings as soon as our new experimental results are available.

---

> > ### Author Response · Authors · 2025-07-17
> > **Update on Additional Experiments**
> >
> > Dear Reviewer 6MbM,
> >
> > Thank you again for your constructive suggestion to test our findings in a more realistic, higher-dimensional setting.
> >
> > We have now completed the suggested experiments using a Bayesian Logistic Regression (BLR) model on the Covertype UCI dataset (55 parameters). We have incorporated the full details of the experimental setup, results, and a discussion into the revised manuscript under a new section, Appendix F.
> >
> > The manuscript has been updated accordingly.
> > We are grateful for the opportunity to improve our work based on your valuable feedback.
> >
> > Sincerely,
> >
> > The Authors

---

### Review · Reviewer_AJy1 · 2025-06-23

**Summary Of Contributions:**

The paper studies the convergence of Stein Variational Gradient Descent in KL divergence. To do so, the authors introduce the $(\epsilon, \delta)$-Approximate Gradient Flow framework which measures the discrepancy between the actual Wasserstein gradient and its approximation used in the algorithm. Then, they demonstrate that SVGD verifies $(c_0, 0)$-AGF in the infinite particle setting, leading to sublinear convergence. Numerical experiments in a Gaussian setting to validate the theoretical results a provided.

**Audience:**

Yes

**Claims And Evidence:**

Yes

**Requested Changes:**

### Typos
* Page 8, lemma 2: in the r.h.s. of the inequality, the sum should be indexed by another symbol than $t$ since $t$ is used outside the sum.

* Page 14: "$\pi\propto \mathrm{exp}^{-L(x)}$"  ->"$\pi\propto \mathrm{exp}^{-V(x)}$"

**Strengths And Weaknesses:**

### Strengths
* The paper is well written and easy to follow.

* The authors show sublinear convergence of SVGD in KL divergence. This result is sound.

* The analysis of SVGD considers a more relaxed setting than the usual convergence in KSD.

* Theory is supported by numerical experiments.

### Weakness

* The code is missing with the submission. Releasing the code would foster reproducibility, even though the experiments are toy.

---

> ### Author Response · Authors · 2025-07-08
> **Author responses**
>
> We thank Reviewer AJy1 for their careful reading and for identifying several typographical errors. We have corrected all of them in the revised manuscript.
>
> **Q1. Typo in Lemma 2 (p. 8)**
>
> **A.** We acknowledge this error. The intention was to sum up to the $(t-1)$-th iteration. We have corrected this in the revised manuscript to avoid ambiguity by using a different index, k, for the summation. The corrected expression in Lemma 2 now reads:
> $\Vert \nabla \log\frac{\mu_t}{\pi}\Vert_{L^2(\mu_t)}\leq \Vert\nabla \log\frac{\mu_0}{\pi}\Vert_{L^2(\mu_0)}+c\sum_{k=0}^{t-1}\gamma_k$.
> This change has been made on page 8 of the revised manuscript.
>
> **Q2. Typo in Appendix B.1 (p. 14)**
>
> **A.** We thank the reviewer for spotting this inconsistency. It is indeed a typo. We have corrected $L(x)$ to $V(x)$ in Appendix B.1 to align with the notation used in the main body of the paper, such as in Assumption 1 on page 3.
>
> **Q3. Regarding the code release.**
>
> **A.** We sincerely thank you for raising the important point of reproducibility. We have provided all source code as supplementary material in the OpenReview submission. We will publicly release the code upon the paper's acceptance.

---

### Review · Reviewer_ksex · 2025-06-25

**Summary Of Contributions:**

This paper addresses theoretical properties of SVGD. Existing techniques consider SVGD as a gradient flow in the RKHS. This perspective makes it difficult to understand the convergence of SVGD in KL divergence under general settings.

A new analytic tool is introduced in this paper that allows the authors to look at he the approximation error between the Wasserstein gradient of the KL divergence and the update rule in SVGD. From this it follows that SVGD exhibits sub-linear convergence in the KL divergence. This is done in the infinite-particle and discrete-time setting.

**Audience:**

Yes

**Claims And Evidence:**

Yes

**Requested Changes:**

I wonder if these changes would be helpful to strengthen the appeal of the paper:
1. Could there be more discussion on whether the analytical tool introduced of epsilon-delta approximate gradient can be applied in broader settings, beyond analysis of SVGD?
2. Perhaps discuss how might a practitioner using SVGD take your results and gain some insight on their implementation of SVGD?

**Strengths And Weaknesses:**

I must admit that this work is quite outside my area of expertise. I only understand the results from a distance. In particular I understand the beginning and end points but have not taken the time to review the theoretical path that connects them.

Strengths:
The paper seems to be technically correct. I like that the paper states quite simply what it sets out to do, and then does it. Too many machine learning papers these days oversell their narrative and are guilty of using mathematics to give only the appearance of rigour. This paper is not guilty of that, which is commendable.

Weaknesses:
As I mentioned, this is outside my area of expertise and thus I can only point out weaknesses on a rather superficial level. The obvious weaknesses is that the analysis is for the infinite-particle setting which is not how SVGD is used in practice. But this weakness is already readily admitted and discussed in the paper.

---

> ### Author Response · Authors · 2025-07-08
> **Author responses**
>
> We thank Reviewer ksex for their thoughtful comments that probe the novelty, practical relevance, and limitations of our work. We address each point below.
>
> **Q1. On the broader applicability and contribution of the $(\epsilon,\delta)$-AGF framework.**
>
> **A.** We argue that our $(\epsilon,\delta)$-AGF framework is a versatile analytical tool with broad applicability.
>
> First, it serves as a unifying lens. As detailed in Section 3.2, it encapsulates existing schemes. For instance, the method of Dong et al. (2022) assumes an error bound of $\Vert g_{\mu_{t}} - \nabla \log \frac{\mu_{t}}{\pi}\Vert_{L^{2}(\mu_t)}^{2} \leq \epsilon \Vert \nabla \log \frac{\mu_{t}}{\pi} \Vert_{L^{2}(\mu_t)}^{2}$, corresponding to a pure relative error case $(\epsilon_t >0, \delta_t=0)$. Conversely, score-based models from Lee et al. (2022; 2023) correspond to a pure absolute error case $(\epsilon_t = 0, \delta_t > 0)$. This shows our framework provides a common language to contrast different algorithms' error structures.
>
> Second, our work introduces a novel analytical approach. Prior studies typically assume an error bound is met. In contrast, our paper derives the $(\epsilon, \delta)$ parameters that SVGD intrinsically achieves, shifting the paradigm from assuming a property to analyzing the algorithm's inherent properties.
>
> Finally, the framework is a valuable diagnostic tool for future research. The decomposition of error into a relative component ($\epsilon_t$) governing the convergence *rate* and an absolute component ($\delta_t$) introducing a final *bias* is powerful. This allows for a more precise diagnosis of algorithmic behavior, providing a structured path for future improvement.
>
> **Q2. On the practical takeaways for SVGD users from Theorem 1 and the numerical experiments.**
>
> **A.** Our theoretical and empirical results offer two key, actionable takeaways for practitioners, which we have clarified in Section 5.
>
> * **Confidence in Convergence and Principled Step-Size:** Our work provides the first KL divergence convergence guarantee for SVGD. This is a significant practical result, offering greater confidence than prior KSD-based guarantees, which do not always imply weak convergence (Gorham & Mackey, 2017). The theoretically-grounded step-size, $\gamma_{t}\leq\mathcal{O}(1/t^{2/3})$, also provides a concrete alternative to difficult heuristic tuning.
>
> * **The Critical Trade-off Between Particle Count and Speed:** Our experiments reveal a critical trade-off: using more particles reduces the final bias but slows down the convergence rate. This arises because with an RBF kernel, more particles lead to exceedingly small eigenvalues that weaken the gradient signal and hinder optimization. This presents practitioners with a direct choice: a faster, more biased solution (fewer particles) versus a slower, more accurate one (more particles), depending on the application's needs.
>
> **Q3. On the limitation of the analysis to the infinite-particle setting.**
>
> **A.** We acknowledge this limitation. We analyze the idealized infinite-particle setting as a necessary first step because the finite-particle case presents significant technical challenges due to particle dependencies, as now detailed in Appendix A.
>
> The core challenge is that with a finite number of correlated particles, ideal assumptions break down. The eigenvectors of the empirical kernel operator are not a complete orthonormal system (CONS), which introduces a projection error. In our framework's language, this manifests as a non-zero absolute bias term, $\delta_t > 0$, preventing the KL divergence from converging to zero. Analyzing this term is extremely difficult because the particle correlation invalidates standard Monte Carlo error analysis.
>
> While our analysis is idealized, it provides the essential AGF framework required to tackle the more challenging finite-particle case, which we identify as a primary direction for future research.

---

### Author Response · Authors · 2025-07-08
**General Introduction from Authors**

We begin by expressing our sincere gratitude to the Area Chair and the anonymous reviewers for their time, effort, and invaluable feedback. The comments are insightful and constructive, and addressing them has significantly improved the clarity, rigor, and overall impact of our manuscript.
We have thoroughly revised the manuscript in light of the feedback. The key modifications include: (1) Correction of all identified typographical errors; (2) Clarifications on the novelty and broader applicability of our proposed $(\epsilon, \delta)$-Approximate Gradient Flow (AGF) framework; (3) An expanded discussion on the practical implications of our theoretical results for users of SVGD; and (4) A more detailed and transparent discussion of the key assumptions and limitations of our work, particularly regarding the infinite-particle setting and Assumption 6.
For the convenience of the reviewers, all significant revisions made to the manuscript are highlighted in magenta. We hope that these revisions, along with our commitment to incorporating further experiments as detailed in our responses, address all reviewer concerns.

**Summary of Revisions:**
- For the convenience of the editor and reviewers, we provide a concise summary of the changes made to the manuscript:
Corrected typographical errors in Lemma 2 and Appendix B.1.
- Added text to Section 3.2 to further clarify the novelty of the AGF framework as a general analytical tool.
- Revised the discussion in Section 5 to more explicitly state the practical trade-off between particle count and convergence speed.
- Revised the discussion in Appendices A and E, summarizing key assumptions (Assumptions 4 and 6), their roles, and their justifications.
- Added a sentence to the Conclusion (Section 6) acknowledging that the empirical estimation of the bias term $\delta_t$​ is a valuable future research direction, building upon the theoretical challenges we identified.

---

### Decision · Action_Editor_XKL4 · 2025-07-29

**Recommendation:** Accept as is

**Audience:**

Yes

**Audience Explanation:**

At least the subset of the TMLR audience who care about the convergence of SVGD will find some interesting new insights in this paper.

**Claims And Evidence:**

Yes

**Claims Explanation:**

The reviewers agree that the theory in the paper is technically sound and provides a novel result on SVGD convergence. The reviewers are also happy to see that the limitations of the work are openly acknowledged by the authors.